# Dataset Diffusion: Diffusion-based Synthetic Dataset Generation for Pixel-Level Semantic Segmentation

**Quang Nguyen**[1,2*]   **Truong Vu**[1*]   **Anh Tran**[1]   **Khoi Nguyen**[1]

[1]VinAI Research, [2]Ho Chi Minh City University of Technology, VNU-HCM

## Abstract

Preparing training data for deep vision models is a labor-intensive task. To address this, generative models have emerged as an effective solution for generating synthetic data. While current generative models produce image-level category labels, we propose a novel method for generating pixel-level semantic segmentation labels using the text-to-image generative model Stable Diffusion (SD). By utilizing the text prompts, cross-attention, and self-attention of SD, we introduce three new techniques: *class-prompt appending*, *class-prompt cross-attention*, and *self-attention exponentiation*. These techniques enable us to generate segmentation maps corresponding to synthetic images. These maps serve as pseudo-labels for training semantic segmenters, eliminating the need for labor-intensive pixel-wise annotation. To account for the imperfections in our pseudo-labels, we incorporate uncertainty regions into the segmentation, allowing us to disregard loss from those regions. We conduct evaluations on two datasets, PASCAL VOC and MSCOCO, and our approach significantly outperforms concurrent work. Our benchmarks and code will be released at https://github.com/VinAIResearch/Dataset-Diffusion.

## 1   Introduction

Semantic segmentation is a fundamental task in computer vision. Its objective is to assign semantic labels to each pixel in an image, making it crucial for applications such as autonomous driving, scene comprehension, and object recognition. However, one of the primary challenges in semantic segmentation is the high cost associated with manual annotation. Annotating large-scale datasets with pixel-level labels is labor-intensive, time-consuming, and requires substantial human effort.

To address this challenge, an alternative strategy involves leveraging generative models to synthesize datasets with pixel-level labels. Past research efforts have utilized Generative Adversarial Networks (GANs) to effectively generate synthetic datasets for semantic segmentation, thereby mitigating the reliance on manual annotation [1–3]. However, GAN models primarily concentrate on object-centric images and have yet to capture the intricate complexities present in real-world scenes.

On the other hand, text-to-image diffusion models have emerged as a promising technique for generating highly realistic images from textual descriptions [4–7]. These models possess unique characteristics that make them well-suited for the generation of semantic segmentation datasets. Firstly, the text prompts used as input to these models can serve as valuable guidance since they explicitly specify the objects to be generated. Secondly, the application of cross and self-attention maps in the image generation process endows these models with informative spatial cues, enabling precise extraction of object positions within the generated images.

By leveraging these characteristics of text-to-image diffusion models, the concurrent works Diffu-Mask [8] and DiffusionSeg [9] effectively generate pairs of synthetic images and corresponding

---

*First two authors contribute equally. The work is done during Quang Nguyen's internship at VinAI Research.

37th Conference on Neural Information Processing Systems (NeurIPS 2023).

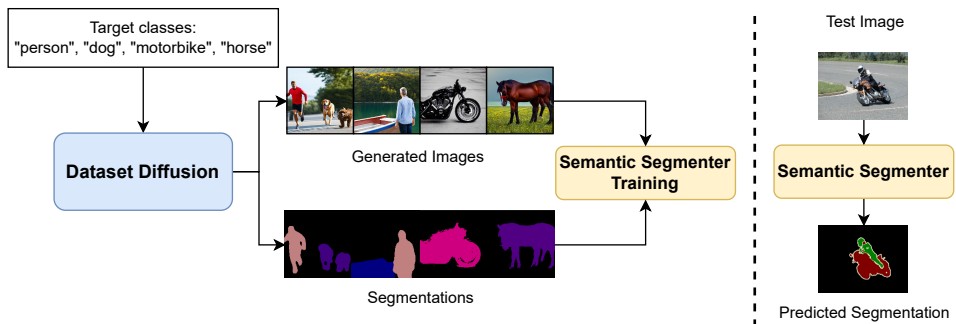

Figure 1: Overview of our Dataset Diffusion for synthetic dataset generation. (**Left**) Given the target classes, our framework generates high-fidelity images with their corresponding pixel-level semantic segmentations. These segmentations serve as pseudo-labels for training a semantic segmenter. (**Right**) The trained semantic segmenter is able to predict the semantic segmentation of a test image.

segmentation masks. DiffuMask achieves this by utilizing straightforward text prompts, such as `"a photo of a [class name] [background description]"`, to generate image and segmentation mask pairs. Meanwhile, DiffusionSeg focuses on creating synthetic datasets that address the challenge of object discovery, which involves identifying salient objects within an image. While these approaches successfully produce images paired with their corresponding segmentation masks, they are currently limited to generating a single object segmentation mask per image.

In this paper, we present Dataset Diffusion, a novel framework for synthesizing high-quality semantic segmentation datasets, as shown in Fig. 1. Our approach focuses on generating realistic images depicting scenes with multiple objects, along with precise segmentation masks. We introduce two techniques: *class-prompt appending*, which encourages diverse object classes in the generated images, and *class-prompt cross-attention*, enabling more precise attention to each object within the scene. We also introduce *self-attention exponentiation*, a simple refinement method using self-attention maps to enhance segmentation quality. Finally, we employ the generated data to train a semantic segmenter using uncertainty-aware segmentation loss and self-training.

To evaluate the quality of the synthesized datasets, we introduce two benchmark datasets: synth-VOC and synth-COCO. These benchmarks utilize two well-established semantic segmentation datasets, namely PASCAL VOC [10] and COCO [11], to standardize the text prompt inputs and ground-truth segmentation evaluation. On the synth-VOC benchmark, Dataset Diffusion achieves an impressive mIoU of 64.8, outperforming DiffuMask [8] by a substantial margin. On the synth-COCO benchmark, the DeepLabV3 model trained on our synthesized dataset achieves noteworthy results of 34.2 in mIoU compared to the model trained on real images with full supervision.

In summary, the contributions of our work are as follows:

- We present a framework that effectively employs a state-of-the-art text-to-image diffusion model to generate synthetic datasets with pixel-level annotations.
- We introduce a simple and effective text prompt design that facilitates the generation of complex and realistic images, closely resembling real-world scenes.
- We propose a straightforward method that utilizes self and cross-attention maps to achieve highly accurate segmentation, thereby improving the quality and reliability of the synthesized datasets.
- We introduce synth-VOC and synth-COCO benchmarks for evaluating the performance of semantic segmentation dataset synthesis.

In the following, Sec. 2 reviews prior work, Sec. 3 describes our proposed framework, and Sec. 4 presents our experimental results. Finally, Sec. 5 concludes with some remarks and discussions.

## 2   Related Work

**Semantic segmentation** is a critical computer vision task that involves classifying each pixel in an image to a specific class label. Popular semantic segmentation approaches include the fully convolutional network (FCN) [12] and its successors, such as DeepLab [13], DeepLabV2 [14], DeepLabv3 [15], DeepLabv3+ [16], UNet [17], SegNet [18], PSPNet [19], and HRNet [20]. Recently,

transformer-based approaches like SETR [21], Segmenter [22], SegFormer [23], and Mask2Former [24] have gained attention for their superior performance over convolution-based approaches. In our framework, we focus on generating synthetic datasets that can be used with any semantic segmenter, so we use DeepLabv3 and Mask2Former as they are commonly used.

**Text-to-image diffusion models** have revolutionized image generation research, moving beyond simple class-conditioned to more complex text-conditioned image generation. Examples include GLIDE [25], Imagen [6], Stable Diffusion (SD) [5], Dall-E [4], eDiff-I [7], and Muse [26]. These models can generate images with multiple objects interacting with each other, more closely resembling real-world images rather than the single object-centric images generated by prior generative models. Our Dataset Diffusion marks a milestone in synthetic dataset generation literature, moving from image-level annotation to pixel-level annotation. We utilize Stable Diffusion [5] in our framework, as it is the only open-sourced pretrained text-to-image diffusion model available at the time of writing.

**Diffusion models for segmentation.** Diffusion models have proven effective for semantic, instance, and panoptic segmentation tasks. These models either use input images to condition the mask-denoising process [27–33], or employ pretrained diffusion models as feature extractors [34–37]. However, they still require ground-truth (GT) segmentation for training. In contrast, our framework utilizes only a pretrained SD to generate semantic segmentation without GT labels.

**Generative Adversarial Networks (GANs) for synthetic segmentation datasets.** GANs have been employed in the generation of synthetic segmentation datasets, as demonstrated in previous works such as [1, 3, 38, 39]. However, these approaches primarily focus on object-centric images, where a single mask is segmented for the salient object or specific parts of common objects like faces, cars, or horses, as exemplified in [2]. In contrast, our framework is designed to generate semantic segmentations for more complex images, where multiple objects interact with each other at the scene level. Furthermore, while some techniques [38, 39] support foreground/background subtraction, and others [1, 3] still require human annotations, our objective is to generate semantic segmentations for multiple object classes in each image without the need for human involvement.

**Diffusion models for synthetic data generation** have been used to improve the performance of image classification [40, 41], domain adaptation for classification [42, 43], and zero/few-shot learning [44–47]. However, these methods produce only image-level annotations as augmentation datasets. In contrast, our framework produces pixel-level annotations, which is considerably more challenging.

Recently, there have been concurrent works [8, 9] that utilize Stable Diffusion (SD) for generating object segmentation without any annotations. However, they focus on segmenting a single object in an image rather than multiple objects. Their text-prompt inputs to SD are simple, usually ``a photo of a [class name]''. The semantic segmenter trained on these annotations can segment multiple objects to some extent. Our framework, on the other hand, employs more complex text prompts where multiple objects can coexist and interact, making it more suitable for the semantic segmentation task in real-world images.

## 3   Dataset Diffusion

**Problem setting:** Our objective is to generate a synthetic dataset $\mathcal{D} = (I_i, S_i)_{i=1}^{N}$, consisting of high-fidelity images $\mathcal{I}$ and pixel-level semantic masks $\mathcal{S}$. These images and masks capture both the semantic and location information of the target classes $\mathcal{C} = \{c_1, c_2, ..., c_K\}$, where $K$ represents the number of classes. The purpose of constructing this dataset is to train a semantic segmenter $\Phi$ without relying on human annotation.

In our approach, we follow a three-step process. Firstly, we prepare relevant text prompts $\mathcal{P}$ containing the target classes (Sec. 3.1). Secondly, using Stable Diffusion (SD) as our model, we generate images $\mathcal{I}_i \in \mathbb{R}^{H \times W \times 3}$ and their corresponding semantic segmentations $\mathcal{S}_i \in \{0, ..., K\}^{H \times W}$, where $0$ represents the background class (Sec. 3.2). These images and segmentations form the synthetic dataset $\mathcal{D}$. Lastly, we train a semantic segmenter $\Phi$ on $\mathcal{D}$ and evaluate its performance on the test set of standard semantic segmentation datasets (Sec. 3.3). It is worth noting that our approach primarily focuses on segmenting common objects in everyday scenes, where the SD model excels, rather than specialized domains like medical or aerial images. The overall framework is depicted in Fig. 2.

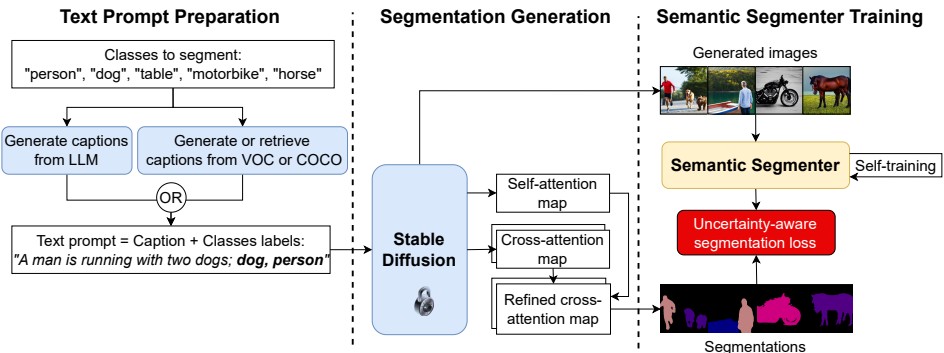

Figure 2: **Three stages of Dataset Diffusion**. In the first stage, the target classes are provided, and text prompts are generated using language models such as ChatGPT [48]. Real captions (for COCO) or image-based captions (for VOC) can also be used for prompt generation to ensure standard evaluation. The text prompts are then augmented with the target class labels to avoid missing objects. In the second stage, given the augmented text prompt, a frozen Stable Diffusion [5] is employed to generate an image and its self- and cross-attention maps. The cross-attention map for each target class is refined using the self-attention map to match the object's shape. Finally, the generated images and corresponding semantic segmentations are used to train a semantic segmenter with uncertainty-aware loss and the self-training technique.

## 3.1 Preparing Text Prompts for Stable Diffusion

To prepare prompts containing a given list of classes for SD, one option is to utilize a large language model (LLM) such as ChatGPT [48] to generate the sentences, similar to the method described in [9]. This approach can be valuable in real-world applications.

However, for evaluating the quality of the synthetic dataset, we need to rely on standard datasets for semantic segmentation like PASCAL VOC [10] or COCO [11] to create standardized benchmarks. In this regard, we propose using the provided or generated captions of the training images in these datasets as the text prompts for SD. This is solely for the purpose of standard benchmarking where the text prompts are fixed, and we do not utilize real images or image-label associations in our synthetic dataset generation. We call these new benchmarks as synth-VOC and synth-COCO.

When using the COCO dataset, we can rely on the provided captions to describe the training images. However, in the case of the PASCAL VOC dataset, which lacks captions, we employ a state-of-the-art image captioner like BLIP [49] to generate captions for each image. However, we encountered several issues with the provided or generated captions. Firstly, the text prompts may not use the exact terms as the target class names $\mathcal{C}$ provided in the dataset. For instance, terms like ''man'' and ''woman'' may be used instead of ''person'', or ''bike'' instead of ''bicycle'', resulting in a mismatch with the target classes. Secondly, many captions do not contain all the classes that are actually present in the images (as illustrated in Fig. 3). This leads to a shortage of text prompts for certain classes, affecting the generation process for those particular classes.

To address the issues, we propose a method that leverages the class labels provided by the datasets. We append the provided (or generated) captions $\mathcal{P}_i$ with the class labels, creating new text prompts $\mathcal{P}'_i$ that explicitly incorporate all the target classes $\mathcal{C}_i = [c_1; \ldots; c_M]$, where $M$ is the number of classes in image $i$. This is achieved through the text appending operation or *class-prompt appending* technique: $\mathcal{P}'_i = [\mathcal{P}_i; \mathcal{C}_i]$. For example, in the case of the left image in Fig. 3, the final text prompt would be ''a photograph of a kitchen inside a house; bottle microwave sink refrigerator''. This ensures that the new text prompts encompass all the target classes, addressing the issue of mismatched or missing class names in the captions.

## 3.2 Generating Segmentation from Self and Cross-attention Maps

We build our segmentation generator on Stable Diffusion (SD) by leveraging its self and cross-attention layers. Given a text prompt $\mathcal{P}'$ first encoded by a text encoder into text embedding $e \in \mathbb{R}^{\Lambda \times d_e}$ with the text length $\Lambda$ and the number of dimensions $d_e$, SD seeks to output the final latent state $z_0 \in \mathbb{R}^{H \times W \times d_z}$, where $H, W, d_z$ are height, width, and number of channels of $z_0$, reflecting the content encoded in $e$ from the initial latent state $z_T \sim \mathcal{N}(0, I)$ after $T$ denoising steps.

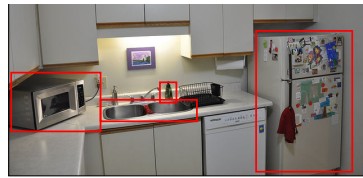
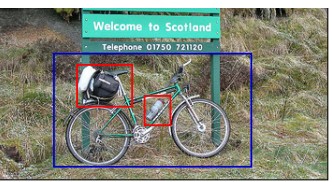
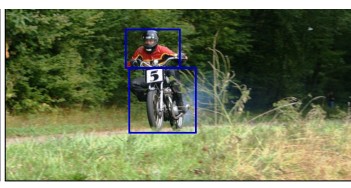
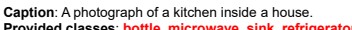

**Caption**: A photograph of a kitchen inside a house. **Provided classes**: bottle, microwave, sink, refrigerator    **Caption**: A **bike** leaning against a sign in Scotland. **Provided classes**: bicycle, backpack, bottle    **Caption**: A **man** riding a dirt **bike** in a forest. **Provided classes**: person, motorcycle

Figure 3: Common issues of using provided (or generated) captions. Red classes are often missing from the captions, resulting in a lack of text prompts for those classes. Blue classes may have different terms used in the captions, causing a discrepancy between the target class names and the text prompts.

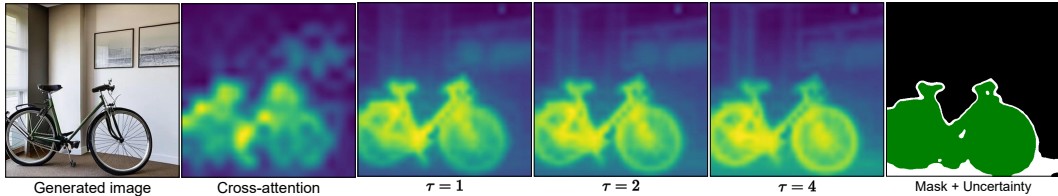

Generated image    Cross-attention    $\tau = 1$    $\tau = 2$    $\tau = 4$    Mask + Uncertainty

Figure 4: Given a text prompt ``A bike is parked in a room; bicycle'', we obtain the generated image, cross-attention map, enhanced cross-attention map by the self-attention with $\tau = \{1, 2, 4\}$ described in the Eq. (3), and mask with uncertainty value (white region) by Eq. (4) and Eq. (5).

At each denoising step $t$, a UNet architecture with $L$ layers of self and cross-attention is used to transform $z_t$ to $z_{t-1}$. In particular, at layer $l$ and time step $t$, the self-attention layer captures the pairwise similarity between positions within a latent state $z_t^l$ in order to enhance the local feature with the global context in $z_t^{l+1}$. In the meantime, the cross-attention layer models the relationship between each position of the latent state $z_t^l$ and each token of the text embedding $e$ so that $z_t^{l+1}$ can express more of the content encoded in $e$.

Formally, the self-attention map $\mathcal{A}_S^{l,t} \in [0,1]^{HW \times HW}$ and cross-attention map $\mathcal{A}_C^{l,t} \in [0,1]^{HW \times \Lambda}$ at layer $l$ and time step $t$ are computed as follows:

$$\mathcal{A}_S^{l,t} = \text{Softmax}\left(\frac{Q_z K_z^\top}{\sqrt{d_l}}\right), \qquad \mathcal{A}_C^{l,t} = \text{Softmax}\left(\frac{Q_z K_e^\top}{\sqrt{d_l}}\right), \qquad (1)$$

where $Q_z, K_z, K_e$ are the query of $z$, key of $z$, and key of $e$, respectively, obtained by linear projections and taken as inputs to the attention mechanisms, and $d_l$ is # features at layer $l$.

Since we only want to obtain the cross-attention map of the class labels $C_i$ of image $i$ for semantic segmentation, we introduce *class-prompt cross-attention* that is similar to cross-attention in Eq. (1) but produced by only taking the softmax over the class name part $C_i$ rather than entire of the text prompt $\mathcal{P}'_i$. In practice, we form a new text prompt $\hat{\mathcal{P}}_i = C_i$ just for the purpose of extracting the cross-attention maps while the original text prompt $\mathcal{P}'_i$ for generating images keeps unchanged. After this, we obtain $\mathcal{A}_C^{l,t} \in [0,1]^{HW \times M}$, where $M$ is the number of classes in the image.

With the observation that using different ranges of timesteps only affects the final result marginally, (provided in Supp.), we average these cross and self-attention maps over layers and timesteps:

$$\mathcal{A}_S = \frac{1}{L \times T} \sum_{l=1}^{L} \sum_{t=0}^{T} \mathcal{A}_S^{l,t}, \qquad \mathcal{A}_C = \frac{1}{L \times T} \sum_{l=1}^{L} \sum_{t=0}^{T} \mathcal{A}_C^{l,t}, \qquad (2)$$

Although the cross-attention maps $\mathcal{A}_C$ already exhibit the location of the target classes in the image, they are still coarse-grained and noisy, as illustrated in Fig. 4. Thus, we propose to use the self-attention map $\mathcal{A}_S$ (as illustrated in Fig. 6 - Left) to enhance $\mathcal{A}_C$ for a more precise object location. This is because the self-attention maps capturing the pairwise correlations among positions within the latent $z_t$ can help propagate the initial cross-attention maps to the highly similar positions, e.g., non-salient parts of the object, thereby enhancing their quality. Therefore, we propose *self-attention exponentiation* where the self-attention map $\mathcal{A}_S$ is powered to $\tau$ before multiplying to the cross-attention map $\mathcal{A}_C$ as:

$$\mathcal{A}_C^* = (\mathcal{A}_S)^\tau \cdot \mathcal{A}_C, \qquad \mathcal{A}_C^* \in [0,1]^{HW \times M}. \qquad (3)$$

Next, we aim to identify two matrices: $\mathcal{V} \in [0,1]^{H \times W}$ representing the objectness value at each location (the higher the objectness, the more likely that location contains an object), and $\mathcal{S} \in \{1, \ldots, M\}^{H \times W}$ indicating which objects in the class labels $C_i$ that each location could be. To obtain those, we perform the pixel-wise $\arg\max$ and $\max$ operator (over the category $M$ dimension):

$$\mathcal{S} = \arg\max_{m} \mathcal{A}_C^{*,m}, \qquad \mathcal{V} = \max_{m} \mathcal{A}_C^{*,m}. \qquad (4)$$

At a location $x$ in the map $\mathcal{V}$, if its value is less than a threshold, one can set its label to the background class 0. However, we find that using a fixed threshold does not work for all images. Instead, we use a lower threshold $\alpha$ for certain background decisions and a higher threshold $\beta$ for certain foreground decisions. Any value that falls inside the range $(\alpha, \beta)$ expresses an uncertain mask prediction with value $U = 255$. That is, the final mask $\bar{\mathcal{S}}$ is illustrated in the last image of Fig. 4 and calculated as:

$$\bar{\mathcal{S}}_x = \begin{cases} 0 & \text{if } \mathcal{V}_x \leq \alpha, \\ U & \text{if } \alpha < \mathcal{V}_x < \beta, \\ S_x & \text{otherwise.} \end{cases} \qquad (5)$$

### 3.3 Training Semantic Segmenter on Generated Segmentation

Given the synthetic images $\mathcal{I}$ and semantic segmentation masks $\bar{\mathcal{S}}$, we train a semantic segmenter $\Phi$ with an uncertainty-aware cross-entropy loss. Specifically, for pixels marked as uncertain, we ignore the loss from those as: $\mathcal{L} = \sum_x \mathbb{1}(\bar{\mathcal{S}}_x \neq U)\mathcal{L}_{\text{CE}}(\hat{\mathcal{S}}_x, \bar{\mathcal{S}}_x)$, where $\mathbb{1}$ is the indication function, $\mathcal{L}_{\text{CE}}$ is the cross entropy loss, and $\hat{\mathcal{S}} = \Phi(\mathcal{I})$ is the predicted segmentation from the generated image $\mathcal{I}$.

We further enhance the segmentation mask $\bar{\mathcal{S}}$ by the self-training technique [50]. That is, after being trained with $\bar{\mathcal{S}}$, the segmenter $\Phi$ makes its own prediction on $\mathcal{I}$ as pseudo labels $\mathcal{S}^*$ without uncertainty value $U$. Finally, the final semantic segmenter $\Phi^*$ is the segmenter $\Phi$ trained again on $\mathcal{S}^*$.

## 4 Experiments

**Datasets:** We evaluate our Dataset Diffusion on two datasets: PASCAL VOC 2012 [10] and COCO 2017 [11]. The PASCAL VOC 2012 dataset has 20 object classes and 1 background class. For standard semantic segmentation evaluation, this dataset is usually augmented with the SBD dataset [51] to have a total of $12,046$ training, $1,449$ validation, and $1,456$ test images. We additionally augment the training images with captions generated from BLIP [49]. The COCO 2017 dataset contains 80 object classes and 1 background class with $118,288$ training and $5K$ validation images, along with provided captions for each image. It is worth noting that we only use the image-level class annotation to form the text prompts as described in Sec. 3.1.

We introduce the set of our prepared text prompts along with the validation set of each dataset as synth-VOC and synth-COCO – the two benchmarks for evaluation of semantic segmentation dataset synthesis. To create a balance synthetic dataset among classes, we generate $2k$ images per object class for PASCAL VOC, resulting in a total of $40k$ image-mask pairs and about $1k$ images per object class for COCO, resulting in a total of $80k$ image-mask pairs. If the number of text prompts associated with a certain class is insufficient, we use more random seeds to generate more images.

**Evaluation metric:** We evaluate the performance of Dataset Diffusion using the mean Intersection over Union (mIoU) metric. The mIoU(%) score measures the overlap between the predicted segmentation masks and the ground truth masks for each class and takes the average across all classes.

**Implementation details:** We build our framework on PyTorch deep learning framework [52] and Stable Diffusion [5] version 2.1-base with $T = 100$ timesteps. We construct the masks using optimal values for $\tau$, $\alpha$, and $\beta$, which are defined in Sec. 6.2. Regarding semantic segmenter, we employ the DeepLabV3 [15] and Mask2Former [24] segmenter implemented in the MMSegmentation framework [53]. We use the AdamW optimizer with a learning rate of $1e^{-4}$ and weight decay of $1e^{-4}$. For other hyper-parameters, we follow standard settings in MMSegmentation.

### 4.1 Main Results

**Quantitative results:** Tab. 1 compares the results of DeepLabV3 [15] and Mask2Former [24] trained on the real training set, a synthetic dataset of DiffuMask [8], and the synthetic dataset of Dataset

Table 1: Comparison in mIoU between training DeepLabV3 [15] and Mask2Former [24] on the real training set, the synthetic dataset of DiffuMask [8], and the synthetic dataset of Dataset Diffusion.

| Segmenter | Backbone | VOC dataset | | | COCO dataset | |
|---|---|---|---|---|---|---|
| | | Training set | Val | Test | Training set | Val |
| DeepLabV3 | ResNet50 | VOC's training (11.5k images) | 77.4 | 75.2 | COCO's training (2017: 118k images) | 48.9 |
| DeepLabV3 | ResNet101 | | 79.9 | 79.8 | | 54.9 |
| Mask2Former | ResNet50 | | 77.3 | 77.2 | | 57.8 |
| Mask2Former | ResNet50 | DiffuMask [8] (60k images) | 57.4 | - | - | - |
| DeepLabV3 | ResNet50 | Dataset Diffusion (40k images) | 61.6 | 59.0 | Dataset Diffusion (80k images) | 32.4 |
| DeepLabV3 | ResNet101 | | 64.8 | 64.6 | | 34.2 |
| Mask2Former | ResNet50 | | 60.2 | 60.5 | | 31.0 |

Table 2: Performance of different text prompt selections. Red: class names, blue: similar terms.

| Method | Example | mIoU (%) |
|---|---|---|
| 1: Simple text prompts | a photo of an aeroplane | 54.7 |
| 2: Captions only | a large white airplane sitting on top of a boat | 50.8 |
| 3: Class labels only | aeroplane boat | 57.4 |
| 4: Simple text prompts + class labels | a photo of an aeroplane; aeroplane boat | 57.6 |
| 5: Caption + class labels | a large white plane sitting on top of a boat; aeroplane boat | **62.0** |

Diffusion. On VOC, our approach yields satisfactory results of 64.8 mIoU when compared to the real training set of 79.9 mIoU. Further, ours outperforms DiffuMask by a large margin of 4.2 mIoU using the same Resnet50 backbone. The detailed IoU of each class is reported in the Supp. Also, Dataset Diffusion achieves a promising result of 34.2 mIoU compared to 54.9 mIoU of real COCO training set. These results demonstrate the effectiveness of Dataset Diffusion, although the gaps with the real dataset are still substantial, i.e., 15 mIoU in VOC and 20 mIoU in COCO. This is due to the fact that the image content of COCO is more complex than that of VOC, reducing the ability of Stable Diffusion to produce images with the same level of complexity. We will discuss more in Sec. 5.

**Qualitative results** on the validation set of VOC are shown in Fig. 5. In Fig. 5a, the synthetic images and their corresponding masks are utilized for training the semantic segmenter. The first two rows (1, 2) serve as excellent examples of successful segmentation, while the last two rows (3, 4) demonstrate failure cases. In certain instances, the self-training technique proves effective in rectifying mis-segmented objects (as seen in rows 2 and 3). However, it can also adversely impact the original masks when dealing with objects of small size (as observed in row 4). In Fig. 5b, our predicted segmentation results on the validation set of VOC exhibit varying outcomes. The first three rows exhibit satisfactory results, with the predicted masks closely aligning with the ground truth. Conversely, the last three rows illustrate failure cases resulting from multiple small objects (row 4) and the presence of intertwined objects (rows 5 and 6).

## 4.2 Ablation Study

We conduct all ablation study experiments on the text prompts described in Sec. 3.1. Additionally, we report the results with 20k images using the initial mask generated by Dataset Diffusion without using the self-training technique or test-time augmentation unless indicated in each experiment.

**Effect of text prompt selection**. Tab. 2 compares different text prompt selection methods. Our *class-prompt appending* technique outperforms the text prompts using captions or class labels only. Specifically, the *class-prompt appending* technique increases the performance by 11.2 and 4.6 mIoU over the "caption-only" and "class-label-only" text prompts, respectively. *Class-prompt appending* also outperforms the simple text prompts by 7.3 mIoU. These results indicate that our text prompt selection method can help SD generate datasets with both diversity and accurate attention.

**Effects of different components** of stage 2 and stage 3 in Fig. 2 on the overall performance are summarized in Tab. 3. Using only cross-attention results in a low performance of 44.8 mIoU as the

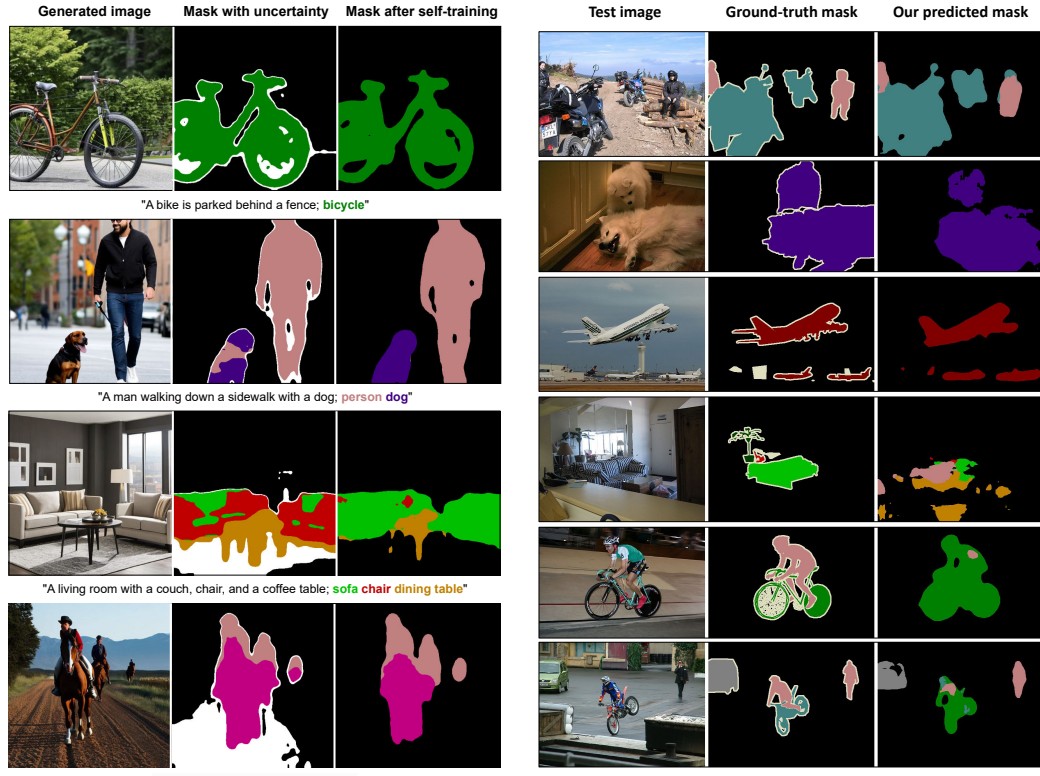

| Generated image | Mask with uncertainty | Mask after self-training | | Test image | Ground-truth mask | Our predicted mask |

"A bike is parked behind a fence; **bicycle**"

"A man walking down a sidewalk with a dog; **person dog**"

"A living room with a couch, chair, and a coffee table; **sofa chair dining table**"

"A man riding a horse; **person horse**"

(a) Our synthetic images and segmentation masks     (b) Segmentation results on VOC's validation set

Figure 5: **(a)** Row 1 (R1) and R2 are successful cases, while R3 and R4 demonstrate failures. Self-training helps correct mis-segmented objects in some cases (R2 and R3) but can harm the original mask for small objects (R4). **(b)** R1 to R3 show accurate results, closely matching the GT. R4 to R6 reveal failure cases due to numerous small objects (R4) and intertwined objects (R5 and R6).

Table 3: Impact of cross-attention, self-attention, uncertainty, self-training, and test time augmentation (TTA) (refer to Sec. 3.2, Sec. 3.3). TTA includes multi-scale and input flipping at test time.

| Cross-attention | Self-attention | Uncertainty | Self-training | TTA | mIoU (%) |
|:---:|:---:|:---:|:---:|:---:|:---:|
| ✓ | | | | | 44.8 |
| ✓ | ✓ | | | | 61.0 |
| ✓ | ✓ | ✓ | | | 62.0 |
| ✓ | ✓ | ✓ | ✓ | | 62.7 |
| ✓ | ✓ | ✓ | ✓ | ✓ | **64.3** |

cross-attention map is coarse and inaccurate (as illustrated in Fig. 4). Using self-attention refinement boosts the performance significantly to 61.0 mIoU. Also, using other techniques like uncertainty-aware loss, self-training, and test time augmentation help improve performance incrementally.

**Effect of different feature scales** used for aggregating self-attention and cross-attention maps is shown in Tab.4. As can be seen, for the cross-attention map, choosing too small and too large feature scales both hurt the performance since the former lacks details while the latter focuses on fine details instead of object shape. For the self-attention map, using the scale of 32 gives slightly better results.

**Hyper-parameters selection for mask generation (Sec. 3.2).** We conduct sensitivity analysis on $\tau$, $\alpha$, and $\beta$ to determine the optimal values in Tab. 5. Tab. 5a shows the results of choosing $\tau$ (with fixed $\alpha = 0.5, \beta = 0.6$) with the best result with $\tau = 4$. A too-large value of $\tau = 5$ decreases the performance as the refined cross-attention map tends to spread out the whole image rather than the object only. Additionally, Tab. 5b exhibits the analysis on the $(\alpha, \beta)$ range given the fixed $\tau = 4$, the range of $(0.5 - 0.6)$ achieves the best performance of 62.0 mIoU.

Table 4: Study on different feature scales

| Cross-attention | Self-attention | |
|---|---|---|
| | 32 | 64 |
| 8 | 39.7 | 38.1 |
| 16 | **62.0** | 59.6 |
| 32 | 52.8 | 50.9 |
| 64 | 35.4 | 31.5 |
| 16, 32 | 59.7 | 57.3 |
| 16, 32, 64 | 59.1 | 57.2 |

Table 5: Hyper-parameters for mask generation.

(a) Analysis of $\tau$ with $\alpha = 0.5$ and $\beta = 0.6$

| $\tau$ | 0 | 1 | 2 | 3 | 4 | 5 |
|---|---|---|---|---|---|---|
| mIoU (%) | 44.8 | 59.5 | 60.5 | 60.2 | **62.0** | 60.5 |

(b) Analysis of $(\alpha, \beta)$ given $\tau = 4$

| $\alpha - \beta$ | 0.4-0.5 | 0.5-0.6 | 0.4-0.6 |
|---|---|---|---|
| mIoU (%) | 59.5 | **62.0** | 60.7 |

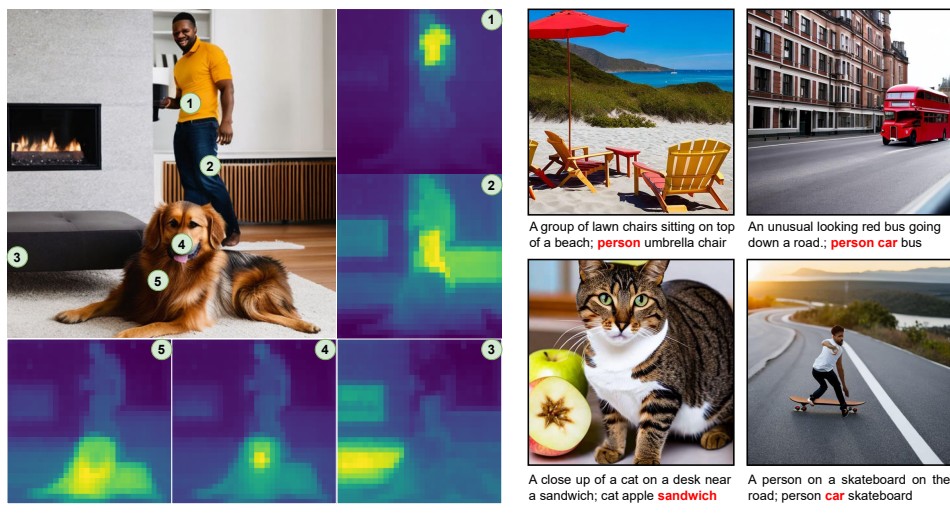

Figure 6: **Left:** Correlation maps at some positions with others, extracted from a self-attention map. **Right:** Failure cases of SD when generating images with multiple objects. Red: classes are missed.

## 5 Discussion and Conclusion

**Limitations:** While our method is effective for generating synthetic datasets, there are some limitations to consider. Our primary reliance on Stable Diffusion [5] for image generation can result in difficulties with producing complex scenes. *First*, when given a text prompt that involves three or more objects, the diffusion model may only produce an image depicting two or three objects as exemplified in Fig. 6 - Right. However, there is ongoing research to improve the quality of the diffusion model and to incorporate stronger guidance, such as layout or box conditions, which shows promise in addressing this issue. *Second*, it is worth noting that in some cases, our Dataset Diffusion may not produce high-quality segmentation masks when objects are closely intertwined, as seen in Fig.5a with the example of a man riding a horse. *Third*, the bias in the LAION-5B dataset, on which Stable Diffusion was trained, may be transferred to the generated dataset. This is the current limitation of Stable Diffusion as it was trained on a large-scale uncurated dataset like LAION-5B. However, there are several studies addressing the bias problem in generative models [54–56] focusing on enhancing fairness and reducing biases in generative models. We believe that these studies and future work on the topic of fairness in GenAI will help to mitigate the bias in the generated images.

**Conclusion:** We have presented our novel framework – Dataset Diffusion – which enables the generation of synthetic semantic segmentation datasets. By leveraging Stable Diffusion, Dataset Diffusion can produce high-quality semantic segmentation and visually realistic images from specified object classes. Throughout our experiments, we have demonstrated the superiority of Dataset Diffusion over the concurrent method, DiffuMask, achieving an impressive mIoU of 64.8 in VOC and 34.2 in COCO. This remarkable advancement paves the way for future research endeavors focused on the creation of large-scale datasets with precise annotations using generative models.

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

# 6 Supplementary Material

In this supplementary material, we first elaborate on our implementation details in Sec. 6.1 Then, we present additional results from our ablation study in Sec. 6.2, focusing on two key aspects: the impact of timesteps on aggregating attention maps and the influence of the number of generated images on the segmentation results. Next, we provide a comprehensive per-class IoU using PASCAL VOC *test* set and COCO2017 *val* set in Sec. 6.3. After that, we present the quantitative results of Dataset Diffusion in other image domains in Sec. 6.4. Finally, we provide more qualitative results in Sec. 6.5.

## 6.1 Elaboration of Implementation Details

**Other text prompt selection methods.** In this part, we discuss how we implement other text prompt selection methods in the first ablation study in the main paper. We note that the difference between these methods and Dataset Diffusion lies in the prompt construction and the cross-attention aggregation. For the 'simple text prompts' method, we construct prompts as ''A photo of a [class name]'' with the cross-attention map at the token "[class name]". For the 'caption only' method, the class labels are not appended to the captions as in Dataset Diffusion with the cross-attention at the class category token. In this way, the terms that do not match exactly with the class names, e.g., "aeroplane" vs. "airplane"), will be ignored. For the "class labels only" method, we use the prompts as ''[class name 1] [class name 2] ...'' with the cross-attention at the class tokens. For compound class names such as "dining table", we take the mean of the cross-attention maps at all positions.

**More on Sec 3.1.** We have observed a major issue with Stable Diffusion. It often fails to generate an adequate number of target classes when multiple object classes are provided in the text prompts, particularly in the case of COCO. To avoid this issue, we introduce a limit on the number of class labels, denoted as $k$. If the number of classes $M$ in an image exceeds this threshold, we select the top-$k$ classes based on their frequency. Then, we generate $k$ simpler text prompts, each containing only one class from $\mathcal{C}_i$ classes of image $i$, ranked by the least frequent classes. The simple text prompt used is ''a photo of a [class name]; [class name]''. This approach ensures that the generated images are more faithful to the provided text prompts, and it facilitates the creation of a more diverse set of text prompts that includes both simple prompts and real captions. This strategy helps to enhance the quality and coverage of the synthetic dataset, mitigating the issue of missing target classes in the generated images.

**Computation details.** We conduct our experiments on NVIDIA A100 40G GPUs. Generating a 40k image dataset with Stable Diffusion V2.1 takes about 30 hours and training the DeepLabV3 for 20k iterations takes about 3 hours on a single GPU.

## 6.2 Additional Ablation Studies

**Study on different ranges of timesteps to aggregate attention maps.** In the main paper, we only experiment with aggregating self-attention and cross-attention maps over all the timesteps. Here, we provide an ablation study in Tab. 6 to show that the variation in timesteps has a minimal impact on the results. As can be seen, averaging all the timesteps yields the best performance, and the maximum decrease is only 0.5 mIoU.

**Study on numbers of generated image-mask pairs in the synthetic dataset.** We evaluate the performance of our semantic segmenter on three numbers of generated images: 10k, 20k, and 40k. The results are shown in Tab. 7. By using a four times larger training dataset (from 10k to 40k), we only gain 1.0% in mIoU. We expect the gain to be smaller when increasing the number of training images further since the system performance is reaching its limit. Because the gain is small compared to the additional computation cost, we do not examine our system when using more than 40k images. We further experiment with the pseudo masks generated using a semantic segmenter well-trained on the real Pascal VOC 2012 dataset to make predictions on our synthetic images. We consider them highly accurate pseudo masks. The results in Tab. 8 show that the performance of the semantic segmenter training on 40k images with these masks yields a 1.4 mIoU gain compared to when training on 10k images, suggesting this small gain comes solely from the mask quality, not from increasing number of generated images.

Table 6: The impact of different ranges of timesteps to aggregate attention maps.

| Timesteps | 100-88 | 88-75 | 75-50 | 50-0 | 100-0 |
|---|---|---|---|---|---|
| mIoU (%) | 61.9 | 61.5 | 61.5 | 61.8 | **62.0** |

Table 7: The impact of different numbers of generated images in the synthetic dataset.

| # images | 10k | 20k | 40k |
|---|---|---|---|
| mIoU (%) | 63.8 | 64.3 | **64.8** |

Table 8: Different # generated images with masked produced by DeepLabV3 trained on real data.

| # images | 10k | 20k | 40k |
|---|---|---|---|
| mIoU (%) | 71.2 | 71.6 | **72.6** |

## 6.3 Detailed Per-class IoU on the VOC and COCO datasets

The detailed per-class IoUs of the COCO 2017 dataset are shown in Tab. 9. Furthermore, in Tab. 10, we report the mIoU on the VOC *test* set of each class when training DeepLabV3 on our synthetic dataset and the Pascal VOC training set. We observe that for the class "tv/monitor", the cross-attention frequently focuses on the boundary of the monitor rather than capturing the entire objects, and the self-attention map cannot help alleviate this issue, resulting in poorly generated masks (as illustrated in Fig. 7). This behavior could explain the significant performance drop of that class compared to the results trained on the real dataset.

## 6.4 Other Image Domains

**Driving Scenes (Tab. 11):** Our synthetic dataset exhibits a performance gap with real datasets (3k real images + manual labeling) of approximately 19 mIoU. This discrepancy parallels those observed in the VOC and COCO datasets, as previously discussed in our main paper. Notably, our synthetic dataset performs comparably with real datasets for specific classes such as bicycle, car, and person. The process of generating images for this dataset is similar to those of VOC and COCO.

**Facial Part Segmentation (Tab. 12):** A performance gap of around 16.5 mIoU (for 5k images) exists between real and synthetic datasets. This disparity is reasonable, considering our approach's zero-shot nature, where training images are generated based on text prompts. Compared to the faces generated in DatasetGAN whose training images are from CelebA-HQ-Mask, the faces generated by Dataset Diffusion are often not well-aligned. However, Dataset Diffusion still competes favorably (78.2 vs 87.0). Furthermore, DatasetGAN requires a number of manual-labeled images (20 in this case) for generating new images and segmentations. Hence, for a fair comparison, we combine our synthetic data with 20 labeled images, and DatasetDiffusion surpasses DatasetGAN (89.9 vs. 87.0). An example of text prompt used is "A portrait photo of a young woman; hair eye nose ear mouth" where hair, eye, nose, ear, mouth are parts' names.

**Satellite/Aerial Images (Tab. 13):** Dataset Diffusion's generated image quality and segmentations do not align with DroneDeploy's standards, leading to a mIoU gap of 37.4. This discrepancy stems from the limited presence of aerial/satellite images in SD's training set (LAION-5B). The text prompts hinder the generation of images that match DroneDeploy's style. The results are presented with the prompt: "An aerial view of a [building, clutter, water, vegetation, ground, car]" and 15k images.

## 6.5 More Qualitative Results

We first demonstrate the qualitative results of the segmenter on the VOC *val* set in Fig. 8 and COCO 2017 *val* set in Fig. 9. Next, in Figs. 10, 11, and 12, we present further qualitative results to illustrate the effectiveness of Dataset Diffusion in generating scenes with different complexities. In Fig. 10, a simple prompt with a single object is depicted, and Dataset Diffusion performs remarkably well

Table 9: COCO 2017's per-class IoU with DeepLabV3 trained on COCO's training set (118k images) and Dataset Diffusion (100k images).

| Class | COCO | Dataset Diffusion | Class | COCO | Dataset Diffusion |
|---|---|---|---|---|---|
| background | 89.1 | 80.9 | wine glass | 48.8 | 29.3 |
| person | 80.6 | 50.5 | cup | 44.8 | 13.7 |
| bicycle | 64.7 | 39.9 | fork | 18.0 | 11.1 |
| car | 53.1 | 35.5 | knife | 19.0 | 6.1 |
| motorcycle | 76.9 | 62.6 | spoon | 15.8 | 7.7 |
| airplane | 78.0 | 62.2 | bowl | 38.0 | 24.4 |
| bus | 77.2 | 64.8 | banana | 62.0 | 49.0 |
| train | 77.2 | 58.7 | apple | 40.1 | 34.9 |
| truck | 52.8 | 36.8 | sandwich | 40.9 | 19.4 |
| boat | 53.1 | 42.5 | orange | 64.4 | 48.4 |
| traffic light | 60.5 | 31.0 | broccoli | 51.0 | 36.8 |
| fire hydrant | 84.6 | 55.9 | carrot | 41.8 | 22.1 |
| stop sign | 90.5 | 42.9 | hot dog | 49.8 | 30.9 |
| parking meter | 68.3 | 51.0 | pizza | 67.8 | 44.1 |
| bench | 46.8 | 27.5 | donut | 62.7 | 48.3 |
| bird | 66.9 | 47.2 | cake | 51.3 | 30.1 |
| cat | 79.9 | 64.0 | chair | 34.3 | 10.5 |
| dog | 74.5 | 46.5 | couch | 52.3 | 29.5 |
| horse | 77.4 | 60.5 | potted plant | 26.1 | 4.5 |
| sheep | 79.7 | 63.1 | bed | 58.3 | 42.0 |
| cow | 75.0 | 63.9 | dining table | 39.6 | 14.6 |
| elephant | 87.4 | 76.6 | toilet | 70.0 | 51.3 |
| bear | 88.4 | 78.1 | tv | 63.8 | 6.3 |
| zebra | 88.2 | 81.8 | laptop | 65.8 | 42.5 |
| giraffe | 82.5 | 77.0 | mouse | 60.3 | 12.7 |
| backpack | 23.1 | 7.5 | remote | 48.6 | 21.2 |
| umbrella | 69.3 | 51.2 | keyboard | 55.5 | 32.6 |
| handbag | 18.9 | 0.04 | cell phone | 57.1 | 35.7 |
| tie | 6.7 | 4.6 | microwave | 59.9 | 39.1 |
| suitcase | 65.8 | 29.8 | oven | 50.1 | 17.8 |
| frisbee | 55.8 | 29.1 | toaster | 0.5 | 6.4 |
| skis | 22.4 | 7.2 | sink | 54.9 | 21.4 |
| snowboard | 48.7 | 21.9 | refrigerator | 70.3 | 20.1 |
| sports ball | 42.1 | 24.5 | book | 37.3 | 23.2 |
| kite | 46.7 | 34.2 | clock | 64.0 | 24.3 |
| baseball bat | 23.1 | 8.3 | vase | 48.0 | 39.4 |
| baseball glove | 59.8 | 2.5 | scissors | 59.1 | 25.1 |
| skateboard | 45.3 | 22.0 | teddy bear | 70.8 | 46.1 |
| surfboard | 60.2 | 43.4 | hair drier | 0.08 | 2.6 |
| tennis racket | 72.6 | 15.7 | toothbrush | 36.6 | 21.9 |
| bottle | 37.5 | 20.1 | **mIoU (%)** | **54.9** | **34.2** |

in this scenario. In Fig. 11, more complex scenes are shown, and Dataset Diffusion still produces acceptable results. However, the limitations of Stable Diffusion become apparent in Fig. 12, where generated scenes are complex but often lack the objects explicitly mentioned in the prompt. This highlights the challenges Stable Diffusion faces when generating scenes with multiple objects.

Table 10: VOC's per-class IoU with DeepLabV3 trained on VOC's training set and Dataset Diffusion.

| Class | VOC | Dataset Diffusion | Class | VOC | Dataset Diffusion |
|-------|-----|-------------------|-------|-----|-------------------|
| aeroplane | 93.6 | 81.6 | diningtable | 58.9 | 42.9 |
| bicycle | 43.8 | 35.8 | dog | 89.7 | 71.8 |
| bird | 93.4 | 73.3 | horse | 93.4 | 78.2 |
| boat | 67.0 | 62.2 | motorbike | 90.9 | 80.6 |
| bottle | 78.5 | 72.6 | person | 87.1 | 70.8 |
| bus | 95.9 | 85.5 | pottedplant | 68.2 | 53.9 |
| car | 90.7 | 64.8 | sheep | 90.9 | 77.8 |
| cat | 94.9 | 78.2 | sofa | 58.7 | 41.8 |
| chair | 36.8 | 21.6 | train | 84.9 | 72.7 |
| cow | 89.4 | 69.2 | tv/monitor | 74.3 | 29.6 |
| | | | **mIoU (%)** | 79.8 | 64.6 |

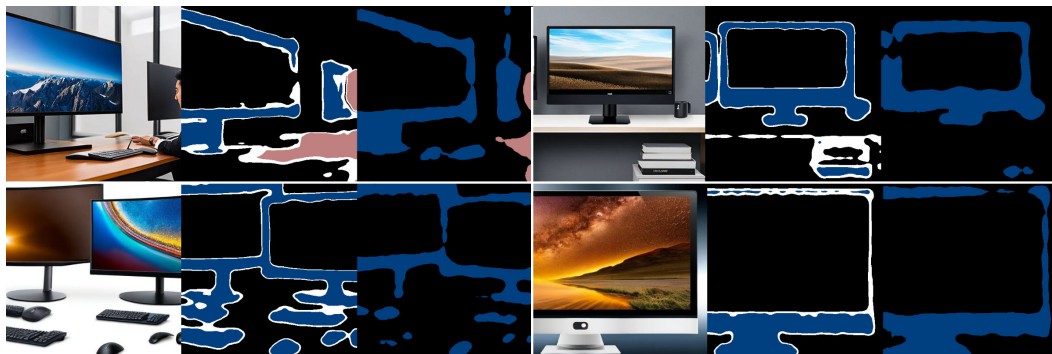

Figure 7: The poorly generated masks for the class "tv/monitor".

Table 11: Cityscapes Object Segmentation Results

| Training set | Bicycle | Bus | Car | Motorbike | Person | Train | mIoU |
|-------------|---------|-----|-----|-----------|--------|-------|------|
| Cityscapes | 78.0 | 89.7 | 95.4 | 69.8 | 82.5 | 84.6 | 83.3 |
| Dataset Diffusion | 72.9 | 57.7 | 93.7 | 39.8 | 82.1 | 39.9 | 64.4 |

Table 12: CelebA-HQ-Mask Facial Part Segmentation Results

| Training set | # Samples | Hair | Eye | Nose | Ear | Mouth | mIoU |
|-------------|-----------|------|-----|------|-----|-------|------|
| CelebA-HQ-Mask | 5k labeled | 98.9 | 98.0 | 99.5 | 77.9 | 99.4 | 94.7 |
| Dataset Diffusion | 5k synthetic | 97.2 | 74.5 | 85.1 | 41.2 | 93.1 | 78.2 |
| DatasetGAN | 5k synthetic + 20 labeled | 97.5 | 95.3 | 96.6 | 48.1 | 95.7 | 87.0 |
| Dataset Diffusion | 5k synthetic + 20 labeled | 97.9 | 95.3 | 97.2 | 61.5 | 97.6 | 89.9 |

Table 13: DroneDeploy Results

| Training set | Building | Clutter | Vegetation | Water | Ground | Car | mIoU |
|-------------|----------|---------|------------|-------|--------|-----|------|
| DroneDeploy | 47.8 | 32.3 | 64.4 | 73.9 | 84.6 | 65.5 | 61.4 |
| Dataset Diffusion | 21.9 | 13.6 | 28.8 | 10.1 | 58.7 | 11.0 | 24.0 |

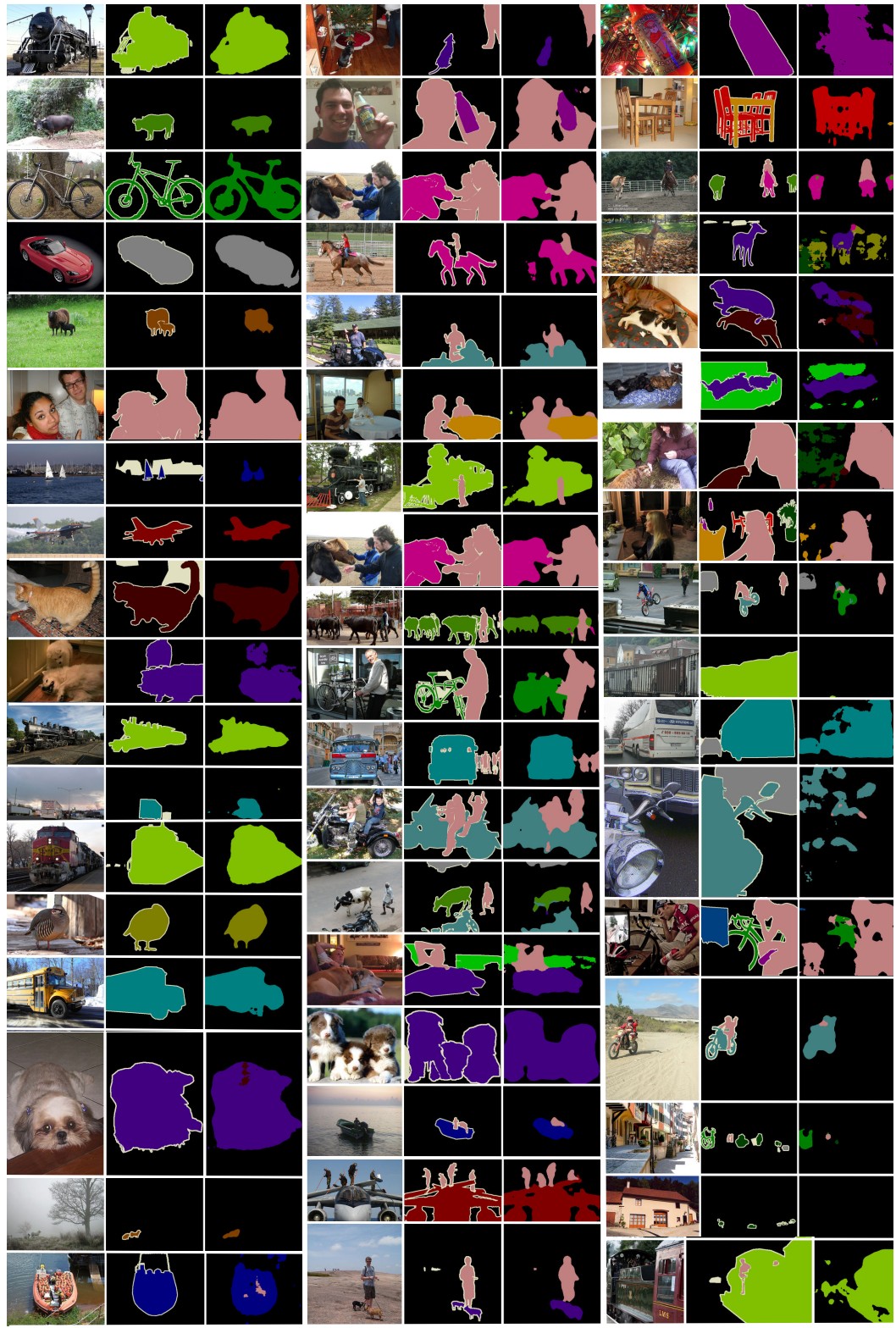

Figure 8: Segmentation results on VOC's *val* set. Three separate columns show examples of simple scenes with a single object, quite complex scenes with multiple objects, and very complex scenes with intertwined objects, respectively. In each example from left to right: test image, ground truth segmentation, and our predicted mask.

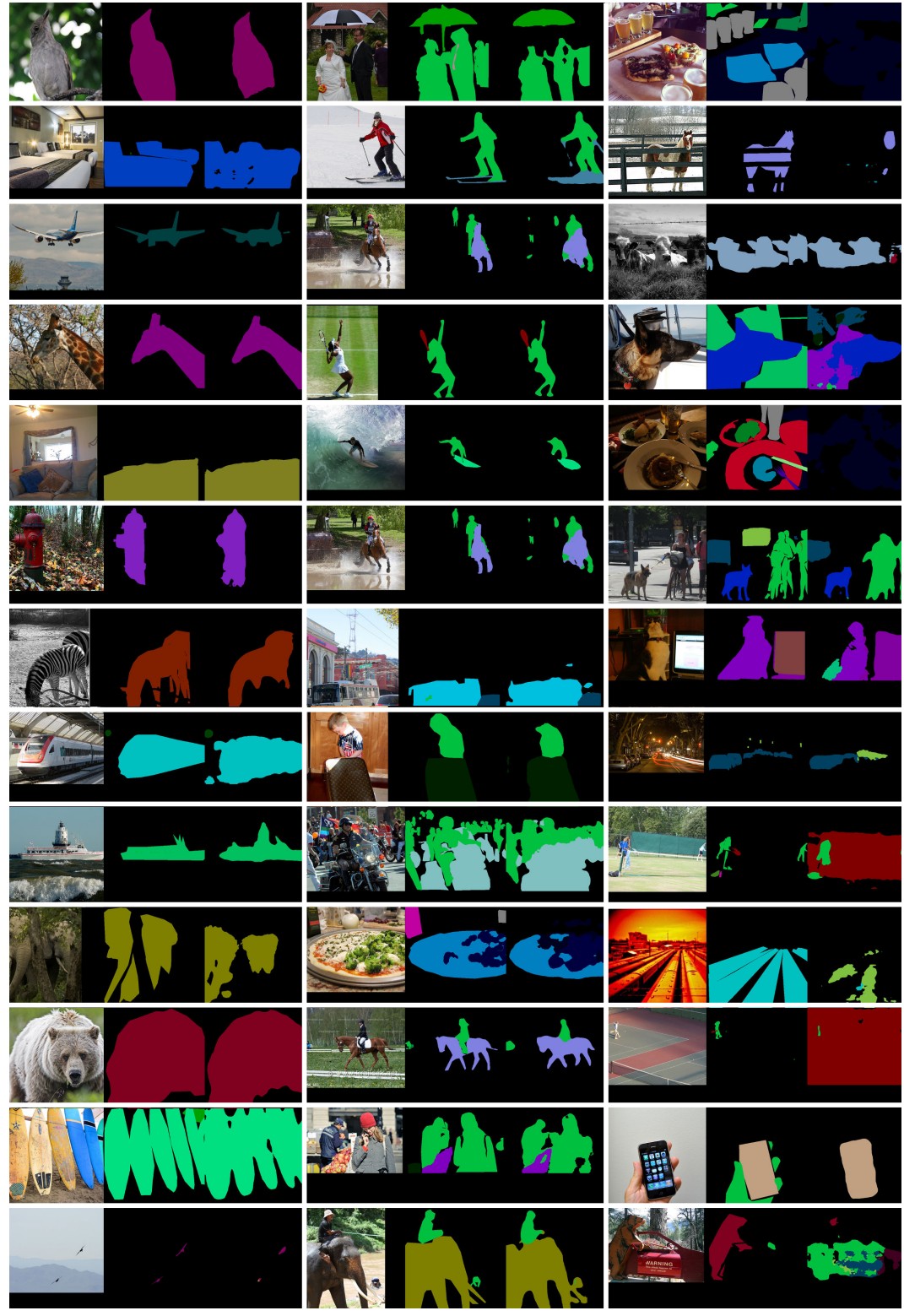

Figure 9: Segmentation results on COCO 2017. Please refer to the caption of Fig. 8 for more details.

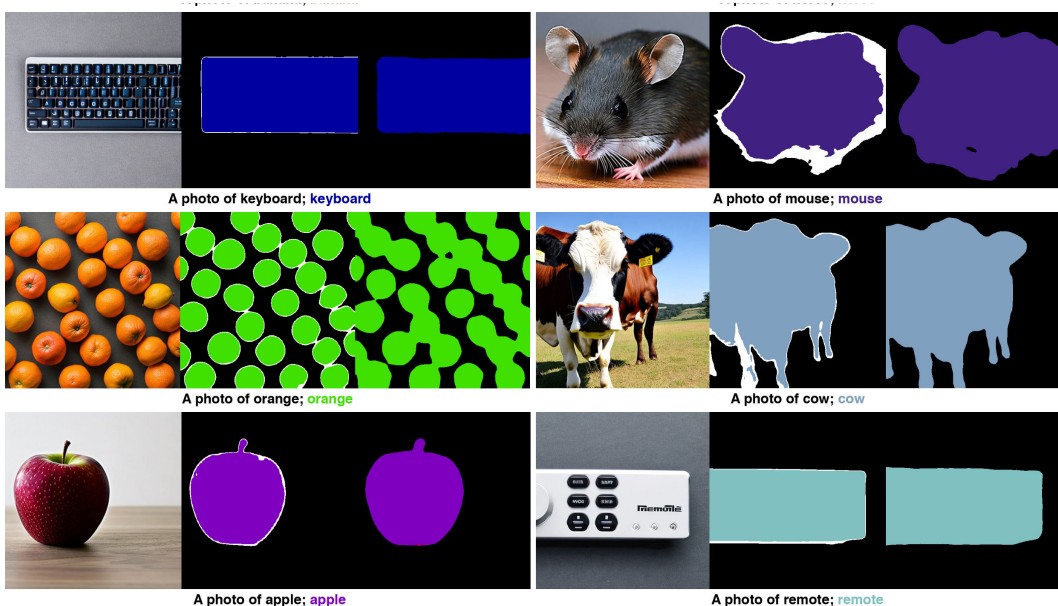

**A photo of keyboard; keyboard**      **A photo of mouse; mouse**

**A photo of orange; orange**      **A photo of cow; cow**

**A photo of apple; apple**      **A photo of remote; remote**

Figure 10: Dataset Diffusion can generate high-quality image-mask pairs with a simple text prompt containing a single object. Each row shows two examples with text prompts underneath. From left to right: generated image, initial mask with uncertain regions in white, and final mask after self-training.

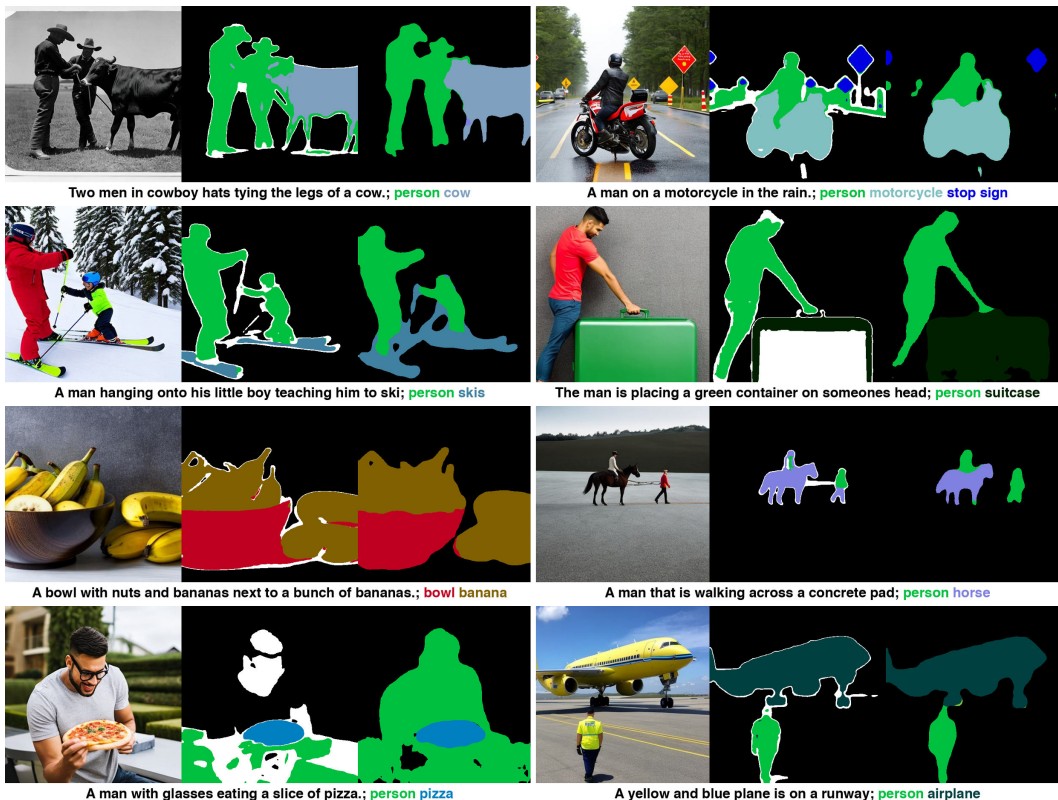

**Two men in cowboy hats tying the legs of a cow.; person cow**      **A man on a motorcycle in the rain.; person motorcycle stop sign**

**A man hanging onto his little boy teaching him to ski; person skis**      **The man is placing a green container on someones head; person suitcase**

**A bowl with nuts and bananas next to a bunch of bananas.; bowl banana**      **A man that is walking across a concrete pad; person horse**

**A man with glasses eating a slice of pizza.; person pizza**      **A yellow and blue plane is on a runway; person airplane**

Figure 11: Dataset Diffusion still works well with some scenes containing multiple objects. Please refer to the caption of Fig. 10 for more details.

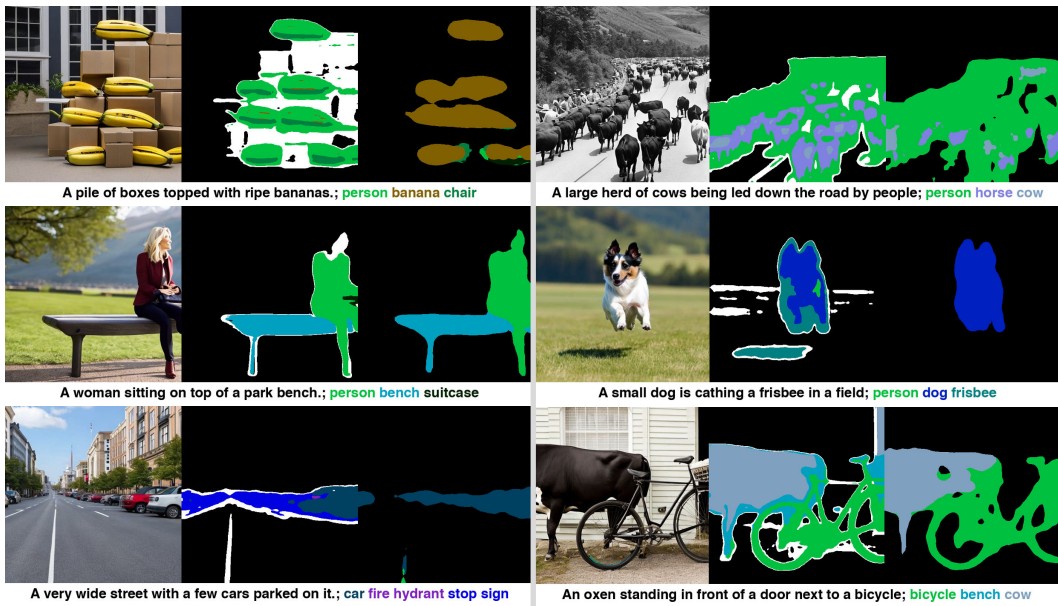

Figure 12: When the prompt becomes excessively complex, Stable Diffusion faces challenges in generating accurate images, i.e., not including all the objects mentioned in the caption, resulting in the absence of these objects' masks. Please refer to the caption of Fig. 10 for more details.

