# OpenReview forum: "Dataset Diffusion: Diffusion-based Synthetic Data Generation for Pixel-Level Semantic Segmentation"
_NeurIPS.cc/2023/Conference — NeurIPS 2023 poster_

### Official Review · Reviewer_wdax · 2023-07-07

**Soundness:** 2 fair
**Presentation:** 2 fair
**Contribution:** 2 fair
**Rating:** 5
**Confidence:** 4

**Summary:**

- The paper introduces a method of generating synthetic training data using Stable Diffusion (SD) for semantic segmentation.

- The class labels are appended to captions which is then used as text prompt to SD to generate synthetic image.

- The segmentation map is generated by refining the cross-attention map (using only the class name as text prompt) using the exponentiated self-attention map.

- The generated segmentation masks with uncertainty regions are used as pseudo-labels for training segmentation models.

- Self-training is performed and the resulting model is evaluated using Test Time Augmentation

**Strengths:**

- Use of self-attention to improve the obtained segmentation mask

- Including uncertain regions in the generated segmentation masks is a good idea as they are extracted from attention maps and hence not high-quality

- Ablation of different components in Table 3 helps understand relative contributions (though more details on the ablation experiment setup would be helpful for the reader)

- Visualization of failure cases

**Weaknesses:**

- L107: “Their text-prompts inputs to SD are simple” - which is inaccurate as the other methods also explore different ways of prompting. Similar to this work, the use of ChatGPT to generate prompts is also explored in [9]

- Finetuning on a small amount of real data improves the performance of [8] significantly and it reaches mIoU higher than training on real data only. Similar experiment here would show how a small amount of real data can be leveraged

- Ablation: simple text prompt with all class labels (i.e. Row1 + Row3 in Table 2)

    - In Table 2, the biggest boost seems to come from using all class labels

    - If simple text prompt with all class labels work well, we could do away with the time consuming extra step of using ChatGPT, BLIP, for generating complex prompts

- The idea of using self-attention for improving the segmentation mask obtained from cross-attention is also explored in [9] -- not referenced in text when discussing

- Missing comparison with [9]

**Questions:**

- is TTA used when evaluating other methods as well?

- visualization of self-attention maps would be good to have

- [8] shows results on open vocabulary segmentation and domain generalization as well, which would make the work more comprehensive

**Limitations:**

Limitations have been discussed in the paper.

- limited to domains/classes for which stable diffusion can generate images

- limited by the complexity of images stable diffusion can generate

- segmentation masks obtained from the method can be noisy/low-quality

- amount of synthetic data possible to generate depends on the inference speed of SD

---

> ### Author Rebuttal · Authors · 2023-08-09
>
> **Q1: L107: “Their text-prompts inputs to SD are simple” - which is inaccurate as the other methods also explore different ways of prompting. Similar to this work, the use of ChatGPT to generate prompts is also explored in [9].**\
> **A1:** What we meant by saying these text-prompt inputs to SD are simple is that only a single object for an image is considered in these prompts, and not simple in the prompt construction. Thank you for pointing out, we will make it clearer in the revised version.
>
> **Q2: Finetuning on a small amount of real data improves the performance of [8] significantly and it reaches mIoU higher than training on real data only. Similar experiment here would show how a small amount of real data can be leveraged.**\
> **A2:** Thanks for your comments. Actually, our finding is opposite as shown in Tab. E of reviewer *2Ket*. Pretraining on synthetic dataset does not help much in the presence of real data. It is not to mention, when real data are available, we can do other strategies to improve the performance with small labeled images such as few-shot learning or semi-supervised learning. Thus, we decided not to involve any real data in our approach and evaluation.
>
>
> **Q3: Ablation: simple text prompt with all class labels (i.e. Row1 + Row3 in Table 2). In Table 2, the biggest boost seems to come from using all class labels. If simple text prompt with all class labels work well, we could do away with the time consuming extra step of using ChatGPT, BLIP, for generating complex prompts**\
> **A3:** Yes, using all class labels can boost the performance significantly compared to just using the simple prompts. However, there is still a very big gap between all class labels alone and combining them with captions (57.4 vs 62.0 in mIoU).
>
>
> **Q4: The idea of using self-attention for improving the segmentation mask obtained from cross-attention is also explored in [9] -- not referenced in text when discussing**\
> **A4:** Thanks for pointing out, we will include in the revised version. DiffusionSeg [9] uses self-attention and cross-attention to generate segmentation mask, however, in a very different way from our approach. In particular, self-attention is employed for constructing the pixel-wise objectness and pair-wise affinity. Given that information, the energy function for each mask can be defined. The mask is then generated by minimizing that function using an off-the-shelf graph cut algorithm. In contrast, our method is much simpler by just multiplying powered self-attention with cross attention to obtain refined cross attention.
>
>
> **Q5: Missing comparison with [9]**\
> **A5:** Please refer to A3 of reviewer *gEQM*.
>
>
> **Q6: is TTA used when evaluating other methods as well?**\
> **A6:** Please refer to A5 of reviewer *2Ket*.
>
>
> **Q7: visualization of self-attention maps would be good to have**\
> **A7:** Please refer to A2 of reviewer *KgKS*.
>
>
> **Q8: [8] shows results on open vocabulary segmentation and domain generalization as well, which would make the work more comprehensive**\
> **A8:** Thank you for your question. We agree that results on open vocabulary segmentation and domain generalization would make our work more comprehensive in terms of comparison with previous work [8].
> For open-vocabulary segmentation, experiments in [8] (Tab. 3) is conducted as follows: training on synthetic dataset of 20 classes of VOC and testing on all 20 classes and split the results into two groups: 15 seen classes and 5 unseen classes. It is not the standard setting where the model is trained on seen classes and tested on unseen classes. For completeness, we provide the results of our approach on the same setting as [8] in Tab. H.
>
>
> **Table H: Zero-shot evaluation**
> | | Segmenter | Seen | Unseen | Harmonic
> | :--------: | :--------: | :--------: | :--------: | :--------:
> | DiffuMask 60k | Mask2Former | 60.8 | 50.4 | 55.1
> | Dataset Diffusion 40k | Mask2Former | **62.7** | **50.9** | **56.2**
>
> For domain generalization or cross-dataset setting, our method is a dataset-agnostic approach, as we only depend on given class names only. Therefore, training with our synthetic data then evaluating on different datasets such as COCO or VOC can be considered as cross-dataset evaluation already. Since [8] has not released the code yet, we conduct the following similar experiments. For each dataset in Tab. I, we extract the subset of six shared classes between VOC and Cityscapes, i.e., bicycle, bus, car, motorbike, person (human + rider), and train. Results in Tab. I suggest that our generated data could achieve competitive results on VOC and Cityscapes dataset, compared to using other cross-domain training data.
>
> **Table I: Cross-dataset evaluation**
> |Train set|Test set| bicycle | bus | car | motorbike | person | train |mIoU|
> | -------- | -------- | :--------: | :--------: | :--------: | :--------: | :--------:| :--------: | :--------:
> | Cityscapes|VOC|61.3|73.2|57.2|69.8|89.9|61.2|68.7
> | Dataset Diffusion|VOC|75.9|92.3|88.8|87.1|93.6|92.1|88.3
> | VOC|Cityscapes|74.5|48.4|94.8|49.8|87.5|19.0 |62.3
> | Dataset Diffusion|Cityscapes|71.2|44.7|92.0|27.8|79.2|5.0 | 53.3

---

> > ### Author Response · Authors · 2023-08-17
> >
> > We hope that our answers address your concerns. If you have any other concerns, please let us know. Thanks!

---

> > ### Comment · Reviewer_wdax · 2023-08-18
> >
> > Thank you for providing detailed answers for all issues raised. I am inclined to raise my rating to borderline accept. I had a few additional clarifications:
> >
> > - For Q3 above, what I meant by combining Row1 + Row3 was provide a combination of simple text prompt with all labels, which in the example of Table 2 would be "A photo of aeroplane and boat". If this performs same as Row4 (using captions + class labels), we could do away with generating captions.
> >
> > - Is it possible to compare to Table 1b from [9] like they do by converting segmentation maps to bounding boxes?

---

> > > ### Author Response · Authors · 2023-08-19
> > >
> > > Thanks for your questions.
> > >
> > > **Regarding Q3**, we have taken your suggestion into consideration. We executed the proposed configuration that includes sample text prompts and class labels (referred to as "simple text prompts with all labels"). We have updated Table 2 in the main paper to the new version, Table 2-new, which now includes an additional row (row 4). The table clearly demonstrates that the performance of the newly proposed prompt aligns with that of class labels alone. This alignment indicates that the captions generated from BLIP significantly contribute to performance enhancement.
> > >
> > > **Table 2-new: Utilizing Simple Text Prompts and Class Labels**
> > > |Method|Example|mIoU|
> > > |--------|--------|:--------:|
> > > |1: Simple Text Prompts with 1 Label|a photo of **an aeroplane**|54.7|
> > > |2: Captions Only|a large white **airplane** sitting on top of a **boat**|50.8|
> > > |3: Class Labels Only|**aeroplane boat**|57.4|
> > > |4: Simple Text Prompts with All Labels|a photo of **an aeroplane** and a **boat**|57.6|
> > > |5: Caption + Class Labels|a large white plane sitting on top of a boat; **aeroplane** **boat**|**62.0**|
> > >
> > > **Question on the conversion of segmentation maps into bounding boxes**, we have followed your recommendation and achieved the reported results in Table M. Upon observation, it is evident from the table that the bounding boxes inferred from our segmentation outperform those inferred from [9] (by approximately 3 points). We intend to incorporate this table into the supplementary material.
> > >
> > > **Table M: Single Object Localization**
> > > |Method|VOC07|VOC12|COCO20K|
> > > | -------- |:--------:|:--------:|:--------:|
> > > |AttentionCut [9]|67.5|70.2|54.9|
> > > |DiffusionSeg [9]|75.2|78.3|63.1|
> > > |Dataset Diffusion|**77.4**|**81.5**|**66.6**|

---

> > > > ### Comment · Reviewer_wdax · 2023-08-20
> > > >
> > > > Thank you for addressing the additional concerns as well. I have revised my rating to borderline accept.

---

> > > > > ### Author Response · Authors · 2023-08-20
> > > > >
> > > > > Thank you!

---

> ### Comment · Area_Chair_Vs85 · 2023-08-17
>
> Dear wdax,
> we would love to hear your thoughts. Did the rebuttal and the other reviews change your mind?

---

### Official Review · Reviewer_2Ket · 2023-07-07

**Soundness:** 3 good
**Presentation:** 3 good
**Contribution:** 3 good
**Rating:** 6
**Confidence:** 4

**Summary:**

The present paper sims to solve the issue of expensive annotation in dense prediction tasks through generating of synthetic images and masks by utilizing frozen a frozen stable diffusion model. First, the authors use the captions extracted from the BLIP or COCO original dataset to generate images by stable diffusion, then combine self-attention and cross-attention maps to produce semantic masks. Finally, the segmenter is supervised by the generated masks with an uncertainty-aware operation, and subsequently self-trained on pseudo labels predicted by the segmenter after first-stage training. This work introduces a simple yet effective method of utilizing text-driven stable diffusion to synthesize images and masks, as opposed to relying heavily on labour-intensive annotations, yielding impressive results on VOC and COCO validation datasets.

**Strengths:**

Strengths:
    1. This work applies the pretrained stable diffusion model to generate synthesized VOC and COCO datasets, which also enables evaluation of the diffusion model’s capability for real-world scenes generation and generalization. This approach could substantially reduce the annotation cost in semantic segmentation task while achieving impressive performance on VOC and COCO validation dataset.
    2. For text prompt generation, this work uses the original captions from COCO dataset and leverages BLIP model to generate captions for VOC, and further introduces a calibration operation to address issues of mismatched and missing class names. Compared with DiffuMask which primarily focus on generating one, this work can generate more complex scene encompassing multiple categories.
    3. For synthesized semantic masks generation, this work ensembles the attention maps of both self-attention and cross-attention. Additionally, they employ uncertainty-aware segmentation loss to alleviate unconfident parameters updates.

**Weaknesses:**

Weaknesses:
	This work proposes a simple but effective strategy to leverage text-driven stable diffusion however lacks novelty. Generating data via diffusion models might introduce computational burdens but holds the potential for effectively addressing the problem of imbalanced data distribution. Moreover, the submitted version lacks of experiments, as detailed in the Questions section.

**Questions:**

Questions:
    1. This work lacks sufficient comparisons of finetuning on real data after training on synthesized data which has been provided in DiffuMask.
    2. The key contribution of this work lies in reducing annotation costs, yet it significantly increases computational costs during training. It would be more convincing to see further comparisons with other generative or self-supervised methods on semantic segmentation tasks in terms of both performance and efficiency.
    3. Is Table 1 a fair comparison with the DiffuMask method? Were self-training and TTA applied in the results of this method?
    4. Since the stable diffusion is totally fixed, how about the results of crossing datasets?
    5. The submission could benefit from meticulous proof-reading and clearer claims. For instance, there are inconsistencies between the values in Table 1 and those reported in the ablation study. Additionally, could the authors specify the type of segmenter and backbone used in the ablation study?

---

> ### Author Rebuttal · Authors · 2023-08-09
>
> **Q1: This work proposes a simple but effective strategy to leverage text-driven stable diffusion however lacks novelty.** \
> **A1:** Our proposed method introduces a simple and unique mask generation process by combining self-attention maps powered to $\tau$ then multiplied with cross-attention maps. To the best of our knowledge, no prior work has explored this particular technique. Therefore, it cannot be considered as lack of novelty.
>
> **Q2: This work lacks sufficient comparisons of finetuning on real data after training on synthesized data which has been provided in DiffuMask**\
> **A2:** We provide the comparisons of fine tuning on real data after training on synthesized data in Tab. E below. We conduct this experiment with Mask2Former segmenter and ResNet-50 backbone. Training directly on 5k real data yielding 77.0 in mIoU (\%), while pre-tranining on synthetic data generated by DiffuMask with 60k images then fine-tuning on 5k real data yields a very marginal result of 77.6 in mIoU (\%).
> Thus, once the real data provided, it is not necesarry to pretrain on synthetic data as proposed in one of the experiments of DiffuMask. Therefore, we decide not to include any real data in our approach or evaluation to merely study the quality of the synthetic data. However, to fulfill the request, we also run Mask2Former pretrained on our synthetic data and obtain slightly better results than that of DiffuMask.
>
>
> **Table E: Comparisons of fine-tuning on real data after pre-training on synthetic data.**
> |Pretrained on synthetic data| Real data | mIoU (\%)
> | :--------: | :--------: | :--------:
> | Not used |VOC 5k| 77.0 |
> |DiffuMask (60k images) |VOC 5k|77.6 |
> |Dataset Diffusion (40k images) |VOC 5k | **78.0** |
>
>
> **Q3: The key contribution of this work lies in reducing annotation costs, yet it significantly increases computational costs during training.**\
> **A3**: Our approach only employs pretrained Stable Diffusion without retraining or finetuning it, so it is a quite affordable and fully-automated process to generate synthetic data. In contrast, real dataset requires  manual data collection and annotation, which is arguably cost more to have same number of annotated images as synthetic dataset. Furthermore, by employing synthetic data, we can effectively address the issue of imbalanced data distribution as suggested by the reviewer.
>
>
> **Q4: It would be more convincing to see further comparisons with other generative or self-supervised methods on semantic segmentation tasks in terms of both performance and efficiency.**\
> **A4:** For other generative models like GAN, please refer to A5 of reviewer *gEQM*. For self-supervised methods, we provide the comparison in Tab. F below. It can be seen that our synthetic data generation approach significantly outperforms self/un-supervised approaches using real images with a very large margin.
>
>
> **Table F: Comparisons with other self-supervised methods on VOC2012 *val set***
> | Method|mIoU (\%)|
> | -------- |:--------:|
> |CLIP$_\text{py}$ ViT-B|54.6
> |MaskDistill+CRF|48.9
> |Leopart|47.2
> |MaskDistill|45.8
> |Dataset Diffusion (ours)|64.8
>
>
> **Q5: Is Table 1 a fair comparison with the DiffuMask method? Were self-training and TTA applied in the results of this method?**\
> **A5**: In Table 1, we ensure a fair comparison with DiffuMask [8] because both methods train the segmenter with only pure synthetic data. In addition, DiffuMask did use the self-training, however, TTA was not mentioned. To address this potential discrepancy and ensure a comprehensive analysis, we present additional results without employing TTA in Tab. G below. As can be seen, we still outperform the version of DiffuMask with a margin of 2.0 mIoU even when not using TTA.
>
> **Table G: Comparative results without using TTA.**
>
> |Training set|Segmenter|Backbone|mIoU
> | -------- | -------- | -------- |:--------:
> | VOC|DeepLabV3|ResNet50|76.2
> | VOC|DeepLabV3|ResNet101|78.7
> | Dataset Diffusion|DeepLabV3|ResNet50|59.9
> | Dataset Diffusion|DeepLabV3|ResNet101|63.1
> | DiffuMask|Mask2Former|ResNet50|57.4
> | Dataset Diffusion|Mask2Former|ResNet50|59.4
>
>
> **Q6: Since the stable diffusion is totally fixed, how about the results of crossing datasets?**\
> **A6**: Our method is a dataset-agnostic approach like DiffuMask, hence we only depends on given class names only. Therefore, training with our synthetic data then evaluating on different datasets such as COCO or VOC can be considered as cross-dataset evaluation already.
> Nevertheless, we also provide the results of cross-dataset evaluation following the setting of DiffuMask [8] for completeness purpose in Tab. I of A8 of reviewer *wdax*.
>
>
> **Q7: The submission could benefit from meticulous proof-reading and clearer claims. For instance, there are inconsistencies between the values in Table 1 and those reported in the ablation study. Additionally, could the authors specify the type of segmenter and backbone used in the ablation study?**\
> **A7:** Thanks! To clarify, the ablation study was conducted using 20k images without self-training and test-time augmentation (TTA) unless explicitly stated in the experiment (L247) whereas the main results presented in Table 1 were achieved by utilizing 40k captions, self-training, and TTA. We used DeepLabV3 segmenter and ResNet101 as backbone for the ablation study.

---

> > ### Comment · Reviewer_2Ket · 2023-08-15
> >
> > I appreciate the author's dedication to providing further clarification and incorporating additional experimental findings. After reading the review from reviewer gEQM, I share the some concern if the proposed method is still useful for non-common scenes.  I will maintain my rating of 'borderline accept'.

---

> > > ### Author Response · Authors · 2023-08-15
> > >
> > > Thanks for your response. We are still working on the new suggested image domains from reviewer gEQM since we have to do the whole experiment again for the new image domains. Stay tuned! We will get you posted.

---

> > > > ### Author Response · Authors · 2023-08-17
> > > >
> > > > Thanks for your patience. We have already addressed Reviewer gEQM's concern in the respective comment above. Please take a look. Hopefully, it also addresses your concern as well.
> > > >
> > > > If you have any further concerns, please let us know.

---

> > > > > ### Author Response · Authors · 2023-08-20
> > > > >
> > > > > Do you have any other concerns or questions that need to discuss? Please let us know. Thanks!

---

> > > > > ### Comment · Reviewer_2Ket · 2023-08-21
> > > > >
> > > > > Thanks for the additional experiments. The outcomes reveal that dataset diffusion achieves reasonable performance with Cityscape and Facial images, but its efficacy somehow diminishes with DroneDeploy datasets. This outcome somewhat validates the apprehension that dataset diffusion might solely excel in common scenarios where ample training data already exists. Nevertheless, this direction remains intriguing and merits deeper investigation. Consequently, I remain inclined to approve this paper and am happy to raise my rating.

---

> > > > > > ### Author Response · Authors · 2023-08-21
> > > > > >
> > > > > > Thanks a lot!

---

### Official Review · Reviewer_KgKS · 2023-07-10

**Soundness:** 3 good
**Presentation:** 3 good
**Contribution:** 3 good
**Rating:** 7
**Confidence:** 5

**Summary:**

The paper addresses the problem of training data preparation for machine learning tasks using generative models. The paper proposes a way to generate pixel-level semantic segmentation dataset using Stable Diffusion. Given a set of target classes, chatGPT produces input text prompts that along with real captions are used to prompt stable diffusion which can sample images. To generate the corresponding segmentation maps, the self-attention map is used to refine the cross-attention map  for each of the target classes. The estimated segmentation masks are then used to train a semantic segmentation network using uncertainty aware loss and self-training methodology.
The efficacy of the method is presented by experiments using PASCAL VOC and COCO dataset. The existing datasets are further enriched by including the captions generated from BLIP. Comparisons are made by training DeepLab V3 and Mask2Former on the real dataset, synthetic dataset from DiffuMask, and the proposed methodology for generating the dataset. Ablations along impact of different design choices is clearly represented in Table 3, along with feature scale in table 4, and hyperparameters for defining the uncertainty in the generated mask.

**Strengths:**

The paper is well presented in terms of motivations first, followed by the technical details defined in detail to reproduce the results. The idea of using Stable Diffusion as a data source is not novel as presented in StableRep and instructPix2Pix, though the urrent paper lays out the method to use its intermediate activations for generating synthetic segmentation masks for training. The comparisons are promising and furthermore the ablations provide enough justifications for the design choice. The method is well described to the extent where it is sufficient for reproducing the results. The presented results are commendatble and in-line with other results on the use of synthetic datas for downstream tasks and representation learning [StableRep]. I am positive about the the use of stable diffusion priors for etracting useful training data. While the results in thsi paper don't beat training with real dataset from COCO and PASCAL VOC, but it the results point to the promise of synthetic data and priros of generative models.


**Weaknesses:**

1. The method requires a fixed set of test classes to be defined beforehand. This is a limitation as the method cannot be used to generate segmentation masks for unseen classes or extend it in an open-vocabulary manner [See OpenSeg].
2. I would like some more discussion on why self-attention has the information to further refine the cross-attention map. It seems like because of the NxN structure of the self-attention map, understanding the process that leads to refinement would be helpful for the reader.
3. How sensitive is the method qualitatively to the choice of the hyperparameters for defining the uncertainty in the generated mask? It would be helpful to see how the uncertainty parameters impact the segmentation masks qualitatively.
4. Since the model is heavily dependent on quality of data in LAION dataset, the bias in the data probably transfers to generated dataset? A section in the main paper or supplementary about some of the known biases in the generated dataset would be helpful.


**Questions:**

1. Can the uncertainty parameters also be conditioned on the input text prompt and the generated segmentation mask using existing segmentation data and its augmentation?
2. Minor suggestion: Please add a few more examples to Figure 6 as there is still some space.

**Limitations:**

Yes. Sufficient discussion was included.

---

> ### Author Rebuttal · Authors · 2023-08-09
>
> **Q1: The method requires a fixed set of test classes to be defined beforehand. This is a limitation as the method cannot be used to generate segmentation masks for unseen classes or extend it in an open-vocabulary manner [See OpenSeg]**.\
> **A1:** Since our method is one of the first works using text-to-image model to generate images and semantic segmentation, we evaluate with the standard semantic segmentations setting and have not considered the unseen classes or open-vocab setting. Moreover, given the synthetic dataset, we can treat it as a real dataset and apply SOTA approaches on unseen classes or open-vocab settings such as OpenSeg. We believe they will totally works fine. However, it is out of the scope of this paper and it might make paper more complicated and cluttered since we focus on the quality of generated synthetic dataset.
>
> **Q2: I would like some more discussion on why self-attention has the information to further refine the cross-attention map. It seems like because of the NxN structure of the self-attention map, understanding the process that leads to refinement would be helpful for the reader.**\
> **A2:** We really appreciate your comment. Cross-attention maps capture the correlation between each position of the latent representation and the tokens of the text embedding. However, the cross-attention map only capture salient parts of the object and ignore non-salient ones. In these cases, the self-attention maps with the ability to capture the pairwise correlations among positions within the latent representation can help propagate the initial cross-attention maps to the highly similar positions, e.g., non-salient parts of the object, thereby enhancing their quality. Also, we provide an illustration of correlation maps extracted from self-attention maps in Fig. 1 in the global attached PDF.
>
> **Q3: How sensitive is the method qualitatively to the choice of the hyperparameters for defining the uncertainty in the generated mask? It would be helpful to see how the uncertainty parameters impact the segmentation masks qualitatively.**\
> **A3**: Thanks! We provide qualitative results with different hyperparameters for defining the uncertainty in generated masks in the Fig. 3 and Fig. 4 of the global attached pdf.
>
> **Q4: Since the model is heavily dependent on quality of data in LAION dataset, the bias in the data probably transfers to generated dataset?**\
> **A4**: We really appreciate your insight. Yes, the bias in the LAION dataset may be transfered to the generated dataset. This is the current limitation of Stable Diffusion as it was trained on a large-scale uncurated dataset like LAION. However, there are several studies addressing the bias problem in generative models:
> + **Seshadri, Preethi, Sameer Singh, and Yanai Elazar. "The Bias Amplification Paradox in Text-to-Image Generation." arXiv preprint arXiv:2308.00755 (2023)**: examines bias amplification in text-to-image generation, focusing on gender biases.
> + **Friedrich, Felix, et al. "Fair diffusion: Instructing text-to-image generation models on fairness." arXiv preprint arXiv:2302.10893 (2023)**: mainly discusses biases related to genders and human behavior.
> + **Su, Xingzhe, et al. "Manifold-Guided Sampling in Diffusion Models for Unbiased Image Generation." arXiv preprint arXiv:2307.08199 (2023)**: proposes a method to estimate the data manifold from the training data. The data manifold is then used as a constraint to guide the sampling process in diffusion models that can mitigate the general data bias.
>
> We believe that with these studies and future work on the topic of fairness in GenAI will help to mitigate the bias in the generated images. We will include this discussion in the revised version.
>
> **Q5: Can the uncertainty parameters also be conditioned on the input text prompt and the generated segmentation mask using existing segmentation data and its augmentation?**\
> **A5**: If we understand your question correctly, you meant that can we have an adaptive threshold for each text prompt rather than a fixed threshold? Yes, we can adapt the threshold with the given text prompt. However, it is not straightforward to do it without significant modification such as a network to predict the adaptive theshold given the text prompts and produced segmentation map. To train the network, it requires a separate dataset which is also non-trivial effort. Therefore, we opt to use a fixed threshold for all text prompts as proposed. However, we still think your idea is intersting and worth more future work on it. Thanks!
>
> **Q6: Minor suggestion: Please add a few more examples to Figure 6 as there is still some space.** \
> **A6**: Thank you for the suggestion.

---

> > ### Author Response · Authors · 2023-08-17
> >
> > Hopefully, our responses addressed your questions and concerns. If you have any further questions, please let us know. Thank you!

---

> ### Comment · Area_Chair_Vs85 · 2023-08-17
>
> Dear KgKS,
> we would love to hear your thoughts. What do you think of the rebuttal and the other reviews?

---

> ### Comment · Reviewer_KgKS · 2023-08-18
>
> Thanks to the authors for addressing all concerns in the review.
>
> - It would be helpful if the section on why self-attention helps in refining the cross-attention can be expanded with a simple experiment to demonstrate this or even build the intuition using the case where this is method is not used.
>
> - Discussion on the bias transfer can be added to the limitation or discussion section so that the reader is aware of this when using the proposed method. Possible ideas of mitigating this bias would be helpful as well.
>
> I will keep the score the same as Accept. Thanks for writing an insightful paper.

---

> > ### Author Response · Authors · 2023-08-19
> >
> > Thank you for your valuable suggestions and encouraging comments. We greatly appreciate your input and will incorporate these discussions into the revised version.
> >
> > Regarding the query about self-attention, we plan to enhance Tab. 5 in the main paper with the updated Table 5-new provided below. Specifically, we will introduce a new column with $\tau=0$, signifying the absence of self-attention for refining cross-attention. The revised table demonstrates that self-attention significantly enhances performance by refining cross-attention, resulting in a notable increase of approximately +15 mIoU. Additionally, we intend to integrate Fig. 1 from the attached global PDF into the main paper. This inclusion will clearly illustrate how self-attention aids in refining cross-attention.
> >
> > **Table 5-new. Absence of Self-Attention Refinement**
> > |$\tau$|0|1|2|3|4|5|
> > | -------- |:--------:|:--------:|:--------:|:--------:|:--------:|:--------:|
> > |mIoU|44.8|59.5|60.5|60.2|**62.0**|60.5

---

### Official Review · Reviewer_gEQM · 2023-07-23

**Soundness:** 2 fair
**Presentation:** 3 good
**Contribution:** 2 fair
**Rating:** 5
**Confidence:** 3

**Summary:**

This work is about automatically generating synthetic data for training semantic segmentation models. Such synthetic data includes realistic input images along with their corresponding ground truth semantic masks. The authors employ text-to-image diffusion models (Stable Diffusion in this work) and propose a specific prompting and ground truth mask generation. The input prompts for Stable Diffusion are a concatenation of an image-level caption (given or generated) as well as the list of category names contained in an image. The ground truth segmentation masks are extracted by combining the self-attention (image tokens) and cross-attention (image and class name text tokens) maps. The evaluation is done on two datasets, Pascal and COCO, and shows superior accuracy compared to one concurrent work (DiffuMask). The ablation study demonstrates the (positive) impact of all aspects of the proposed framework.

**Strengths:**

- The general problem of generating synthetic training data for segmentation models from vision-language models is interesting and promising.
- The proposed prompting for Stable Diffusion is simple but effective.
- The paper is well written and easily comprehensible. The figures give a good overview and also explain each of the proposed aspects of the work well.
- All aspects of the proposed solution are evaluated in the ablation study.

**Weaknesses:**

- L123: Unfortunately, I think creating synthetic data is most useful for exactly those applications that are not based on everyday scenes. There are plenty of datasets with everyday scenes already available, like Pascal, COCO or ADE20K.
- Figure 2 and L133: Doesn't using captions from VOC and COCO bias the whole system and make it unfair when evaluating on those same dataset? Other methods that do not leverage these captions may be at a disadvantage. Using only generated captions would be fine, though. It would have been great to see the difference in final accuracy for ground truth captions and generated captions.
- Table 1: The comparison to prior work is lacking. There is only a single comparison to a prior/concurrent work. Did DiffuMask [8] also use self-training? And why is there no comparison to DiffusionSeg [9]? Why is there no comparison to GAN-based methods for synthetic dataset generation? Is there a way to use the evaluation protocol from [8, 9] for a comparison?

**Post-rebuttal:**
I acknowledge that I read all reviews and the author's feedback. The author's feedback clarified several of my concerns and I raised my rating to "borderline accept". I'd love to see a discussion about the limitations (and potentially some supporting numbers) in the paper, as discussed in the comments.

**Questions:**

**Questions:**
- The feature scales in Table 4 look sensitive. Do you get a similar conclusion for other datasets?

**Suggestions:**
- L35: I think it would be useful to the reader if some high-level context is given on how DiffuMask (or the proposed method) achieves generation of segmentation masks.
- L147: Shouldn't $M$ rather be $M_i$ if it is image-dependent?
- L148: Does concatenating two strings really need a method name like "text appending operation" or "class-prompt appending technique"? It's a very basic operation.

**Limitations:**

Yes, limitations have been addressed in the paper.

---

> ### Author Rebuttal · Authors · 2023-08-09
>
> **Q1: I think creating synthetic data is most useful for exactly those applications that are not based on everyday scenes. There are plenty of datasets with everyday scenes already available, like Pascal, COCO, or ADE20K.** \
> **A1:** Thanks for your question. We want to clarify that our approach support all kinds of image domains a text-to-image model, i.e., Stable Diffusion can generate. In the paper, we present the experiments on everyday scenes like VOC and COCO since we only have the ground-truth semantic segmentation from this image domain for evaluation. Furthermore, even for the everyday image domain, there are still some cases such as imbalance data distribution, long-tail distribution of object categories, or rare classes that these datasets cannot represent but our approach can generate synthetic datasets for these cases very well as exemplified by some examples of rare classes in Fig. 2 of the global attached pdf. We also select 5 rare classes in the LVIS dataset to train a semantic segmenter on limited real training images (30 images in total) and a sufficient synthetic set (250 images with 50 images each) and report the results in Tab. A below. Our synthetic data generation is an effective tool for tackling rare classes.
>
> **Table A: Results on five rare classes of the LVIS dataset**
> |  |horse buggy | garbage | hippopotamus | dice | mallet | mIoU
> |:-------- |:--------: |:--------:|:--------:|:--------:|:--------:|:--------:|
> | Training set of LVIS | **84.1** | 19.6 | 39.7 | 42.9 | 21.9 | 41.6
> | Dataset Diffusion | 82.6 | **68.9** | **63.8** | **64.2** | **43.0** | **64.5**
>
> **Q2: Figure 2 and L133: Doesn't using captions from VOC and COCO bias the whole system and make it unfair when evaluating on those same dataset? Other methods that do not leverage these captions may be at a disadvantage. Using only generated captions would be fine, though. It would have been great to see the difference in final accuracy for ground truth captions and generated captions.**\
> **A2:** Thanks! We consider two kinds of generated captions: from an image captioner and from an LLM like ChatGPT. For the former, we already tested it in VOC experiments since we do not have GT captions for this dataset. For the latter, we show the results on VOC in Tab. B below with generated 40k prompts using ChatGPT. We observe no considerable differences in performance between the two generated captions.
> An example of the text prompt generated by ChatGPT is "A yellow compact car driving through a city next to buses; car, bus". On COCO, we are the first to report the results of using synthetic data to train a semantic segmenter, thus, no comparison is available. It's worth noting that our proposed evaluation protocol based on the image caption is aimed to be a standard benchmark for future work in this direction. That is, we primarily focus on better segmentation techniques rather than on better prompt engineering with ChatGPT.
>
> **Table B: Results in mIoU (%) on the VOC test set with caption generated by BLIP and text prompt generated by ChatGPT with DeepLabV3 without TTA and self-training**
>
> | Source captions | ResNet50 | ResNet101 |
> |:-------- |:--------:|:--------:|
> | Captions from BLIP | 58.5 | 62.2
> | Prompts from ChatGPT | 58.3 | 61.2
>
>
> **Q3: Table 1: The comparison to prior work is lacking. There is only a single comparison to a prior/concurrent work. Did DiffuMask [8] also use self-training? Why is there no comparison to DiffusionSeg [9]? Is there a way to use the evaluation protocol from [8, 9] for a comparison?** \
> **A3:** It's worth noting that we are one of the first works exploring the direction of generating a synthetic dataset for semantic segmentation using a text-to-image diffusion model. Other concurrent works like [8, 9] are published on Arxiv and have not published their code yet. Among them, only DiffuMask [8] can be directly compared to ours and we follow their evaluation setting on VOC. [8] also reported results with the self-training strategy on VOC. On the other hand, Diffusionseg [9] focuses on saliency detection instead and did not release code Therefore, we cannot compare our approach with them.
>
> **Q4: Why is there no comparison to GAN-based methods for synthetic dataset generation?** \
> **A4:** GAN-based approaches like DatasetGAN [1], BigDatasetGAN [3], [38], and [39] are tested on different tasks such as part segmentation and keypoint detection in [1], and single object segmentation or saliency detection in [3], [38], and [39]. Also, it's not trivial to modify their codes to work with multiclass semantic segmentation like in VOC or COCO. Therefore, we cannot compare directly with these GAN-based approaches. However, we still try our best to adapt [39] to work with VOC and report the results in Tab. C, where 2 classes "person" and "horse" are excluded. The results from the table demonstrate the superior performance of our approach (a diffusion-based method) over the GAN-based approach for multiclass semantic segmentation.
>
> **Table C: Comparison results with [39] on 18 VOC classes**
> |Method|mIoU|
> |:--------:| :--------:|
> |[39] (GAN-based)|20.3|
> |Dataset Diffusion|**62.5**|
>
> **Q5: The feature scales in Table 4 look sensitive. Do you get a similar conclusion for other datasets?** \
> **A5:** Yes, we also get a similar conclusion on COCO with 20k captions. We conduct this ablation study with the same setting as Tab. 4. We also achieve the best results when using a cross-attention map at 16 resolution and a self-attention map at 32 resolution as shown in Tab. D below. Hence, the results are consistent among VOC and COCO.
>
> **Table D: Study on different feature scales on COCO**
> |Cross-attention|Self 32|Self 64|
> | :--------: | :--------: | :--------: |
> | 8        |16.2    | 15.5
> | 16       |**25.1**    | 23.9
> | 32       |21.6    | 21.7
> | 64       |15.7    | 15.2
> | 16,32    |24.2    | 23.5
> | 16,32,64 |23.7    | 23.8
>
> **Suggestions**: Thanks so much, we will revise the main paper accordingly

---

> > ### Comment · Reviewer_gEQM · 2023-08-12
> >
> > **Q1:**
> >
> > > In the paper, we present the experiments on everyday scenes like VOC and COCO since we only have the ground-truth semantic segmentation from this image domain for evaluation
> >
> > There exist many segmentation datasets that could be used for evaluation I guess. [PapersWithCodes](https://paperswithcode.com/datasets?task=semantic-segmentation&page=1) provides a long list, including datasets for domains like satellite images, facial part segmentation, driving scenes, etc.
> >
> > **Q4:**
> >
> > > GAN-based approaches like DatasetGAN [1], BigDatasetGAN [3], [38], and [39] are tested on different tasks such as part segmentation
> >
> > You could also evaluate your method on part segmentation I guess, no?

---

> > > ### Author Response · Authors · 2023-08-14
> > >
> > > Thanks for your suggestions. We are working on your suggested image domains and soon have the results reported. We really appreciate your patience.

---

> > > > ### Author Response · Authors · 2023-08-17
> > > >
> > > > We sincerely appreciate your patience and wish to express our gratitude for your invaluable suggestions.
> > > >
> > > > We would like to address a key aspect of our approach that primarily excels within the realm of image generation, specifically leveraging a text-to-image diffusion model known as Stable Diffusion (SD). It's crucial to note that our approach demonstrates its greatest efficacy when applied to image domains that are integral to the training set of SD, notably the LAION 5B dataset. While this limitation may be evident, we firmly believe that this will be rectified in forthcoming iterations of SD, as it expands to encompass a wider list of image domains. We will include this insight in the main paper.
> > > >
> > > > **Regarding Q1**: We present our approach's results across three distinct image domains: driving scenes (evaluated on Cityscapes), facial part segmentation (tested on CelebA-HQ-Mask), and satellite images (assessed on DroneDeploy), as detailed in Tables J, K, and L, respectively.
> > > >
> > > > **Table J: Cityscapes Object Segmentation Results**
> > > > |Training Set| Bicycle | Bus | Car | Motorbike | Person | Train | mIoU |
> > > > | ----------- | -------- | ------- | --- | ------ | ---------- | ----- | ---- |
> > > > | Cityscapes [a] | 78.0 | 89.7 | 95.4 | 69.8 | 82.5 | 84.6 | 83.3 |
> > > > | Dataset Diffusion | 72.9 | 57.7 | 93.7 | 39.8 | 82.1 | 39.9 | 64.4 |
> > > >
> > > > **Table K: CelebA-HQ-Mask Facial Part Segmentation Results**
> > > > |Training Set| # Samples | Hair | Eye | Nose | Ear | Mouth | mIoU |
> > > > | ----------- | -------- | ------- | --- | ------ | ---------- | ----- | ---- |
> > > > | CelebA-HQ-Mask [b] | 5k labeled | 98.9 | 98.0 | 99.5 | 77.9 | 99.4 | 94.7 |
> > > > | Dataset Diffusion | 5k synthetic | 97.2 | 74.5 | 85.1 | 41.2 | 93.1 | 78.2 |
> > > > | DatasetGAN [1] | 5k synthetic + 20 labeled | 97.5 | 95.3 | 96.6 | 48.1 | 97.5 | 87.0 |
> > > > | Dataset Diffusion | 5k synthetic + 20 labeled | 97.9 | 95.3 | 97.2 | 61.5 | 97.6 | 89.9 |
> > > >
> > > > **Table L: DroneDeploy Results**
> > > > |Training Set| Building | Clutter | Vegetation | Water | Ground | Car | mIoU |
> > > > | ----------- | -------- | ------- | --- | ------ | ---------- | ----- | ---- |
> > > > | DroneDeploy [c] | 47.8 | 32.3 | 64.4 | 73.9 | 84.6 | 65.5 | 61.4 |
> > > > | Dataset Diffusion | 21.9 | 13.6 | 28.8 | 10.1 | 58.7 | 11.0 | 24.0 |
> > > >
> > > > From these tables, it is evident that Dataset Diffusion generates synthetic datasets of commendable quality. Allow us to elaborate on the key findings:
> > > >
> > > > 1. **Driving Scenes** (Table J): Our synthetic dataset exhibits a performance gap with real datasets (3k real images + manual labeling) of approximately 19 mIoU. This discrepancy parallels those observed in VOC and COCO datasets, as previously discussed in our main paper. Notably, our synthetic dataset performs comparably with real datasets for specific classes such as bicycle, car, and person. The process of generating images for this dataset is smilar to those of VOC and COCO.
> > > > 2. **Facial Part Segmentation** (Table K): A performance gap of around 16.5 mIoU (for 5k images) exists between real and synthetic datasets. This disparity is reasonable, considering our approach's zero-shot nature, where training images are generated based on text prompts. Compared to the faces generated in DatasetGAN whose training images are from CelebA-HQ-Mask, the faces generated by Dataset Diffusion are often not well-aligned. However, Dataset Diffusion still competes favorably (78.2 vs 87.0). Furthermore, DatasetGAN requires a number of manual-labeled images (20 in this case) for generating new images and segmentations. Hence, for a fair comparison, we combine our synthetic data with 20 labeled images, and DatasetDiffusion surpasses DatasetGAN (89.9 vs. 87.0). An example of text prompt used is "A portrait photo of a young woman; hair eye nose ear mouth" where hair, eye, nose, ear, mouth are parts' names.
> > > > 3. **Satellite/Aerial Images** (Table L): Dataset Diffusion's generated image quality and segmentations do not align with DroneDeploy's standards, leading to a mIoU gap of 37.4. This discrepancy stems from the limited representation of aerial/satellite images in SD's training set (LAION-5B). The text prompts hinder the generation of images that match DroneDeploy's style. The results are presented with the prompt: "An aerial view of a [building, clutter, water, vegetation, ground, car]" and 15k images.
> > > >
> > > > [a] M. Cordts, et al., "The Cityscapes Dataset for Semantic Urban Scene Understanding," Proc. of IEEE CVPR, 2016.\
> > > > [b] Z. Liu, et al., "Deep learning face attributes in the wild," ICCV, December 2015.\
> > > > [c] M. R. Heffels, J. Vanschoren, "Aerial imagery pixel-level segmentation," arXiv:2012.02024, 2020.
> > > >
> > > > **For Q4**: Certainly, we have indeed evaluated our approach's performance on the CelebA-HQ-Mask facial part segmentation dataset. This evaluation is presented in Table K, and further elaborated upon in discussion point 2 above. Note that, only DatasetGAN [1] reported the results on Facial Part Segmentation task while others [3, 38, 39] did not.

---

> > > > > ### Author Response · Authors · 2023-08-20
> > > > >
> > > > > We hope our answers address your concerns well and can change your mind.
> > > > >
> > > > > Fundamentally, our approach can handle various image domains from everyday images such as VOC, and COCO, to less common image domains such as human faces or driving scenes which make up a good chunk of Stable Diffusion's training set. Also, our approach is working for rare image domains like satellite/aerial images but not as well as other image domains. Furthermore, even for the everyday image domain, our approach still stands out in the case of class imbalance, long-tail distribution, or rare object classes. Our experiments support these claims.
> > > > >
> > > > > One cool thing about our Dataset Diffusion is that it doesn't rely on labor-intensive tasks like manually collecting images or labeling pixel-wise annotations. Instead, it leans on a pre-trained foundational model like Stable Diffusion, which is already available. This sets us apart from methods using real datasets. We've got a strong belief that our approach will inspire new directions in future research. These directions can potentially enhance the quality of synthetic data, working towards bridging the gap in performance between synthetic and real datasets.
> > > > >
> > > > > Regarding your concern about comparing with DiffusionSeg [9], we follow the suggestion from reviewer *wdax* to compare the results of single object localization deduced from our segmentation mask. You can find these outcomes detailed in Tab. M in the response directed to reviewer *wdax*. Notably, our approach exhibits a significant performance superiority over [9], with a notable gap of 3 points over three datasets.
> > > > >
> > > > > If you've got more questions or thoughts, please don't hesitate to let us know - and do it soon. Thanks a lot!

---

> ### Comment · Reviewer_gEQM · 2023-08-21
>
> Thanks for doing the additional experiments. These certainly help address my concerns. And I think they also do point out the limitations of the proposed method better. Domains that are not well handled by the underlying diffusion method, won't work that well. Kind of obvious, but still important I think. I'll raise my rating, but I'd love to see a discussion about the limitations (and potentially some supporting numbers) in the paper.

---

> > ### Author Response · Authors · 2023-08-21
> >
> > We would like to express our gratitude for your valuable comments to help point out the image domain that our approach does not work well. As promised earlier, we will add the discussion and support numbers to the paper. Thanks so much!

---

### Author Rebuttal · Authors · 2023-08-10

We thank all reviewers for their positive feedback. All the reviewers recognize the problem of generating synthetic training data for semantic segmentation from text-to-image diffusion model is interesting and promising. In addition, they also find that our paper is well-written and easy to follow. Moreover, reviewers *2Ket* and *wdax* praise the proposed approach incorporating self-attention and cross-attention maps to generate the semantic masks is simple but effective, can generate more complex semantic segmentation than DiffuMask. They also agree that employing uncertainty-aware segmentation is a good idea to alleviate the imperfect semantic masks. Furthermore, reviewers *gEQM*, *KgKS*, and *wdax* also compliment our comprehensive ablation study. Also, *gEQM* and *2Ket* also appreciate our proposed simple but effective prompting for Stable Diffusion. Finally, *wdax* comments that we have a good visualization of failure cases. Below, we address other comments point by point.

---

### Decision · Program_Chairs · 2023-09-21

**Decision:**

Accept (poster)

**Comment:**

The reviewers found the paper to be well written, include comprehensive controlled experiments, and presenting a simple & novel approach. They were mostly concerned about the dependence on the initial dataset for training the underlying diffusion model and how it might affect performance on rare classes. Another concern was the missing comparison to GAN-based approaches. Concerns by the reviewers were sufficiently addressed in the extensive rebuttal. The reviewers and the AC appreciated the additional experiments and results. Adding them in combination with a discussion of limitations (i.e. performance depends on how well SD captures the respective domain) will strengthen the paper substantially and push the paper over the acceptance threshold.